# Genes with epigenetic alterations in human pancreatic islets impact mitochondrial function, insulin secretion, and type 2 diabetes

**Tina Rönn** [1], **Jones K. Ofori** [1], **Alexander Perfilyev** [1], **Alexander Hamilton** [1,2], **Karolina Pircs** [3,4,5], **Fabian Eichelmann** [6,7], **Sonia Garcia-Calzon**[1,8], **Alexandros Karagiannopoulos** [1], **Hans Stenlund** [9], **Anna Wendt**[1], **Petr Volkov**[1], **Matthias B. Schulze**[6,7,10], **Hindrik Mulder**[1], **Lena Eliasson** [1], **Sabrina Ruhrmann**[1], **Karl Bacos** [1,11] & **Charlotte Ling** [1,11] ✉

Epigenetic dysregulation may influence disease progression. Here we explore whether epigenetic alterations in human pancreatic islets impact insulin secretion and type 2 diabetes (T2D). In islets, 5,584 DNA methylation sites exhibit alterations in T2D cases versus controls and are associated with HbA1c in individuals not diagnosed with T2D. T2D-associated methylation changes are found in enhancers and regions bound by β-cell-specific transcription factors and associated with reduced expression of e.g. *CABLES1*, *FOXP1*, *GABRA2*, *GLR1A*, *RHOT1*, and *TBC1D4*. We find RHOT1 (MIRO1) to be a key regulator of insulin secretion in human islets. *Rhot1*-deficiency in β-cells leads to reduced insulin secretion, ATP/ADP ratio, mitochondrial mass, Ca²⁺, and respiration. Regulators of mitochondrial dynamics and metabolites, including L-proline, glycine, GABA, and carnitines, are altered in *Rhot1*-deficient β-cells. Islets from diabetic GK rats present Rhot1-deficiency. Finally, *RHOT1*methylation in blood is associated with future T2D. Together, individuals with T2D exhibit epigenetic alterations linked to mitochondrial dysfunction in pancreatic islets.

Type 2 diabetes (T2D), the fastest growing non-infectious disease worldwide[1], is characterized by hyperglycemia caused by insufficient insulin release from pancreatic islets, often in combination with insulin resistance. Research dissecting the mechanisms that perturb islet insulin secretion is needed to better understand T2D pathogenesis and find new drug targets.

DNA methylation, a dynamic epigenetic mechanism, regulates gene activity and cell function. Epigenetic dysregulation can lead to

[1]Department of Clinical Sciences Malmö, Lund University Diabetes Centre, Scania University Hospital, Malmö, Sweden. [2]Department of Biology, University of Copenhagen, København, Denmark. [3]Laboratory of Molecular Neurogenetics, Department of Experimental Medical Science, Wallenberg Neuroscience Center and Lund Stem Cell Center, Lund University, Lund, Sweden. [4]HCEMM-Su, Neurobiology and Neurodegenerative Diseases Research Group, Budapest, Hungary. [5]Institute of Translational Medicine, Semmelweis University, Budapest, Hungary. [6]Department of Molecular Epidemiology, German Institute of Human Nutrition Potsdam-Rehbruecke, Nuthetal, Germany. [7]German Center for Diabetes Research, München-Neuherberg, Germany. [8]Department of Food Science and Physiology, Centre for Nutrition Research, University of Navarra, Pamplona, Spain. [9]Swedish Metabolomics Centre, Umeå University, Umeå, Sweden. [10]Institute of Nutritional Science, University of Potsdam, Nuthetal, Germany. [11]These authors contributed equally: Karl Bacos, Charlotte Ling. ✉e-mail: charlotte.ling@med.lu.se

disease, and there is evidence of associations between the epigenome and T2D[2,3]. Smaller cohorts have shown that DNA methylation is different in human pancreatic islets from T2D cases versus non-diabetic controls[4–6]. Functional experiments in cultured β-cells of identified candidate genes showing both differential DNA methylation and expression in islets from donors with T2D (e.g., *CDKN1A*, *PDE7B*, *PARK2*, and *SOCS2*) further linked epigenetic dysregulation in pancreatic islets to impaired insulin secretion[4,6]. Moreover, increased islet DNA methylation of T2D candidate genes, such as *INS*, *PDX1*, *GLP1R*, and *PPARGC1A*, has also been associated with reduced expression of said genes and abrogated insulin secretion, further supporting a key role of epigenetic dysregulation in pancreatic islets from individuals with diabetes[7–10]. However, though these case-control studies identified differential DNA methylation of numerous genes in pancreatic islets from donors with T2D, they did not establish whether the identified epigenetic alterations predispose an individual to disease. Therefore, studies testing whether epigenetics predispose to T2D and whether epigenetic alterations can be found already in islets from individuals at risk for T2D are desirable. Such studies could provide support that epigenetic alterations in pancreatic islets may cause diabetes rather than being a consequence of the disease. In addition, if epigenetic alterations predispose an individual to diabetes, this information can be used for preventive care, delaying disease onset and progression, and reducing long-term complications and patient suffering. Moreover, larger case-control studies that robustly link epigenetics to T2D are warranted.

Our aims were, firstly, to perform an epigenome-wide association study in pancreatic islets from a larger T2D case-control cohort and test whether the T2D-associated epigenetic alterations are also linearly associated with HbA1c using islets from individuals not previously diagnosed with diabetes (Fig. 1a). Secondly, we tested whether the identified epigenetic alterations are associated with future T2D based on DNA methylation in blood samples from a prospective cohort (Fig. 1a). We integrated our findings with islet data on expression, open chromatin regions (OCRs), transcription factor (TF) binding occupancy, and histone modifications. Ultimately, we silenced identified candidates in human islets and clonal β-cells to dissect their effects on cell metabolism and insulin secretion and validated the findings in a rodent model of T2D (Fig. 1a).

## Results

### Altered DNA methylation in pancreatic islets from individuals with T2D

To better understand the epigenetic basis of T2D, DNA methylation of 816,888 sites was analyzed using EPIC arrays in the Islet T2D case-control cohort including 25 donors diagnosed with T2D and 75 non-diabetic controls (Table 1). This revealed 31,806 sites with differential methylation in T2D cases versus controls (Supplementary Data 1, false discovery rate [FDR] ≤5%, $q < 0.05$); 77% had lower methylation in islets from T2D cases than in islets from controls. After adjusting for cell composition[11], methylation of 24,546 of the 31,806 sites was associated with T2D ($p = 4.2 \times 10^{-20}$–$4.9 \times 10^{-2}$; Supplementary Data 1). The largest absolute methylation difference was 21.1%, at cg15132295 in *DNAH17* (Fig. 1b). Lower *DNAH17* methylation has been linked to hepatic cancer[12]. For increased methylation in T2D cases, the largest absolute difference was 19.5% (cg09972436 in *LCE3C*, Fig. 1c), a gene linked to autoimmune diseases[13]. We also calculated the fold change in DNA methylation, as the impact of absolute differences may depend on whether basal methylation levels are low or high. We found percentage differences up to 80%, meaning that the proportion of cells being methylated at a specific site was almost doubled in one group (Supplementary Data 1). This was true for a site in *DSCR3*, with 12.9% methylation in controls and 23.3% methylation in T2D cases (Fig. 1d). *DSCR3* methylation is a potential biomarker for preeclampsia[14]. A heatmap of the 100 sites with largest differences in DNA methylation

between T2D cases and controls visualizes the consistency between individuals in each group (Fig. 1e).

To further explore this comprehensive data and its biological context, we used the methylGSA package to perform a pathway analysis on the whole methylation data set[15]. Seventy-two KEGG pathways were associated with T2D ($q < 0.05$), including the MAPK, Calcium, and Insulin signaling pathways, Type II diabetes mellitus, Pancreatic secretion, and Purine metabolism (Supplementary Data 2). Figure 1f visualizes some key pathways, supporting a strong link between islet methylation and T2D.

### T2D-associated methylation changes occur before T2D diagnosis

To understand whether islet methylation is associated with elevated blood glucose, a predictive risk factor for T2D, we studied the association between HbA1c levels and DNA methylation in islets from 114 individuals not previously diagnosed with T2D (Islet HbA1c cohort; Table 1). We found 18,422 sites where methylation was associated with HbA1c ($q < 0.05$; Supplementary Data 3), 60% negatively. After adjusting for cell composition[11], methylation of 10,938 of the 18,422 sites was associated with HbA1c ($p = 8.6 \times 10^{-13}$–$4.9 \times 10^{-2}$; Supplementary Data 3). Importantly, methylation of 5584 sites was associated with both T2D and HbA1c. These were all in concordance, i.e., methylation sites with a positive association with HbA1c exhibited higher methylation in islets from individuals with T2D, and vice versa, suggesting that HbA1c-associated methylation levels may predict future T2D. Of the 5584 sites, 4047 associated negatively with HbA1c and were methylated at lower levels in T2D cases, whereas 1537 had a positive association between methylation and HbA1c and increased methylation in T2D islets (Fig. 1g, Supplementary Data 4).

A pathway analysis on the whole methylation data set based on HbA1c associations revealed 53 enriched pathways ($q < 0.05$). The result resembled that of T2D associations, and 47 pathways overlap with pathways enriched in T2D, including Type II diabetes mellitus and Pancreatic secretion (Supplementary Data 5, Fig. 1f). Overall, the data support that epigenetic alterations in islets predispose an individual to T2D.

### Validation of islet EPIC array data

To technically validate the EPIC array methylation data, we designed pyrosequencing assays for four regions covering methylation sites associated with both T2D and HbA1c (Supplementary Methods). Validation was performed in islets (24 T2D cases, 59 controls; Supplementary Data 6) using the same statistical model as in the EPIC analysis. All pyrosequencing assays confirmed differences in methylation (Fig. 2a–d). For *RHOT1* (also known as *MIRO1*), we analyzed the methylation of two additional sites (Supplementary Methods) 159 and 162 bp downstream from cg10339923, which supported reduced methylation in T2D islets (Fig. 2e, f).

We also compared the current EPIC results with a previous study using 450 K arrays and islets of 15 T2D cases and 34 controls[4]. There was a small overlap of islet donors with the current study (5 cases, 18 controls). Importantly, of sites reported by Dayeh et al. to have altered methylation in islets from T2D cases ($q < 0.05$ and Δβ ≥ 5%), we could replicate 813 sites with consistent differences between cases and controls (Supplementary Data 7), including sites annotated to *CACNA1C*, *CDKN1A*, *GLP1R*, *HDAC4*, *HDAC7*, *IL6R*, *KCNQ1*, *PDE7B*, and *THADA* (Supplementary Fig. 1a–i). In both studies, 98.5% of these 813 sites were hypomethylated in islets from T2D cases versus controls (Supplementary Data 7).

Next, we compared our data to a previous whole genome bisulfite sequencing (WGBS) analysis of DNA methylation in islets from six T2D cases and eight controls[6]. We found that 1297 of the methylation sites associated with T2D and 712 of the sites associated with HbA1c in the current study are within differentially methylated regions (DMRs)

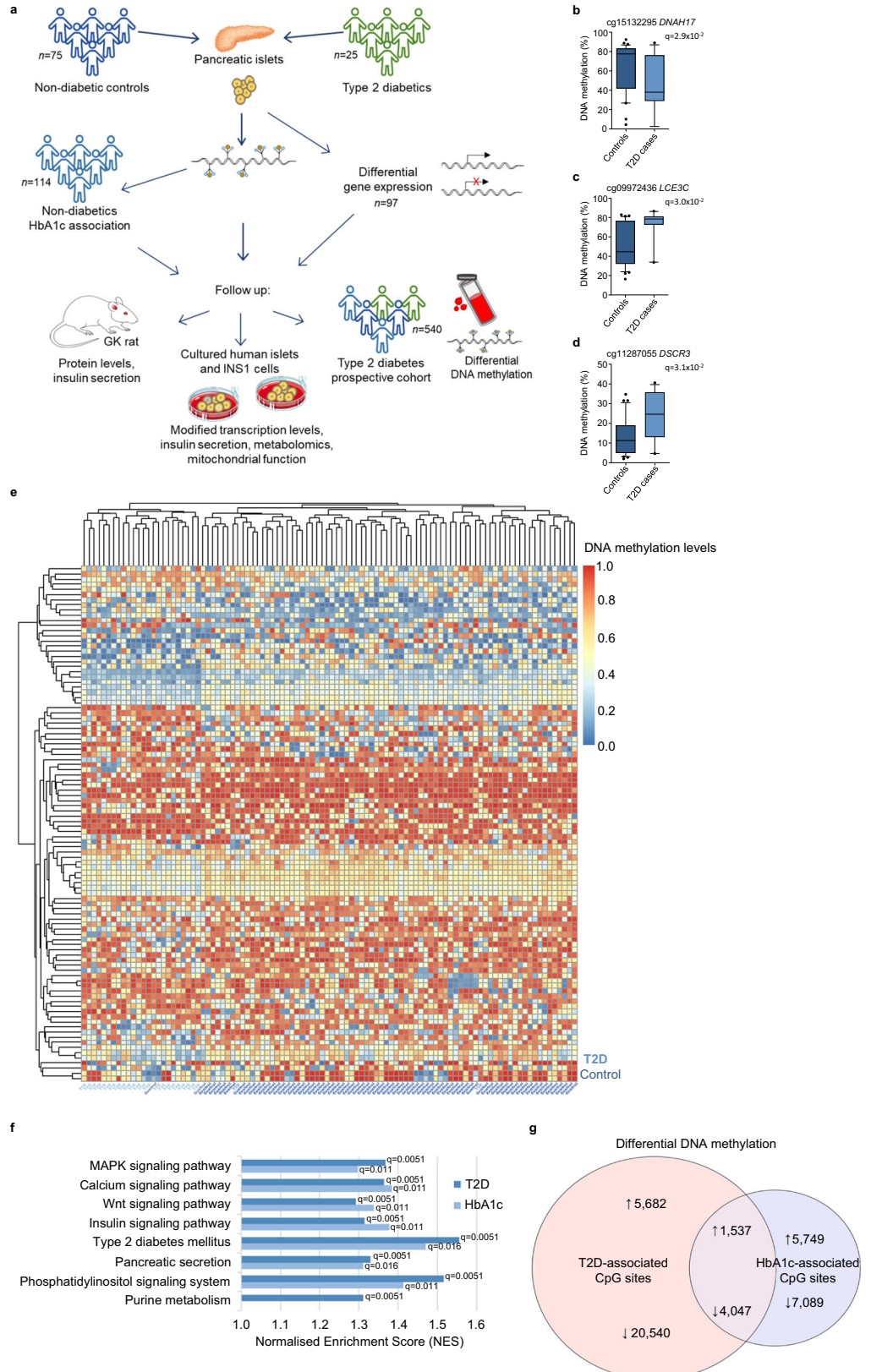

**f**

MAPK signaling pathway
Calcium signaling pathway
Wnt signaling pathway
Insulin signaling pathway
Type 2 diabetes mellitus
Pancreatic secretion
Phosphatidylinositol signaling system
Purine metabolism

Normalised Enrichment Score (NES)

■ T2D
■ HbA1c

**g** Differential DNA methylation

T2D-associated CpG sites
↑5,682
↑1,537
↑5,749
HbA1c-associated CpG sites
↓4,047
↓ 20,540
↓7,089

identified by WGBS (Supplementary Data 7). DMRs contain consecutive sites (range 3–164), and reported methylation differences are averages for the whole DMR (≥5%). These include sites annotated to *DNMT3A*, *GLP1R*, *HDAC4*, *IL6R*, *KCNQ1*, *PDX1*, *TCF7L2*, *THADA*, and *WFS1* (Supplementary Fig. 1c, d, f–g, i–m). 72% of the 1297 sites were hypomethylated in islets from T2D cases versus controls (Supplementary Data 7).

## Altered DNA methylation in open chromatin regions

OCRs typically mark active DNA regions, including enhancers, promoters, and intragenic regions. Altered chromatin accessibility may affect expression, thereby contributing to disease[2]. We previously used ATAC-seq to analyze the chromatin structure in islets from donors with and without T2D[16] and found 79,255 ATAC-seq peaks,

**Fig. 1 | Differential DNA methylation in the *Islet T2D case-control* and *Islet HbA1c* cohorts. a** Study design. EPIC array DNA methylation in islets of 25 T2D cases and 75 controls. RNA-seq data were available for 25 T2D cases and 72 controls. Islet DNA methylation was associated with HbA1c in 114 individuals not diagnosed with T2D. Methylation was also measured in blood from a prospective T2D cohort, and candidate genes were functionally validated in human islets, INS-1 832/13 β-cells, and GK rats. Analyses detected decreased methylation of *DNAH17* (**b**) and increased methylation of *LCE3C* (**c**) and *DSCR3* (**d**) in T2D cases vs. controls, shown as absolute methylation levels (%). *q*-values are based on a linear regression analysis adjusted for age, BMI, sex, days in culture, and islet purity, after correction for multiple testing (FDR). *n* = 75 controls and 25 T2D cases. Box plots indicate median (middle line), 25th, 75th percentile (box) and 5th and 95th percentile (whiskers) as well as outliers (single points). Source data for **b**–**d** are provided as a Source Data file. **e** Hierarchical clustering heatmap showing the DNA methylation pattern for the 100 CpG sites with the largest differences in the Islet T2D case-control cohort. Samples are shown on the x-axis, and methylation sites on the y-axis. **f** Selected KEGG pathways associated with T2D based on genome-wide DNA methylation data from the Islet T2D case-control cohort (dark blue; *q*-values < 0.05 after adjustment for multiple testing, based on robust rank aggregation, followed by gene set enrichment analysis). Most pathways were also associated with HbA1c based on genome-wide DNA methylation data from the Islet HbA1c cohort (light blue; *q*-values < 0.05 after adjustment for multiple testing, based on robust rank aggregation, followed by gene set enrichment analysis). **g** Overlap of sites with differential DNA methylation in the Islet T2D case-control cohort and Islets HbA1c cohort. ↑: increased methylation in T2D cases vs. controls, positive association with HbA1c. ↓: lower methylation in T2D cases vs. controls, negative association with HbA1c. Numbers are the number of associated methylation sites (*q* < 0.05). Panel **a** was partly generated using Servier Medical Art, provided by Servier, licensed under a Creative Commons Attribution 3.0 unported license.

representing OCRs, including 1078 with different prevalence in islets from T2D cases versus controls. Notably, 10,183 of the T2D-associated methylation sites in the current study reside in islet OCRs (Supplementary Data 8), indicating variable DNA methylation levels in regions of active genes. Of these, 2043 methylation sites were also associated with HbA1c in islets (Supplementary Data 8). Moreover, 158 of our T2D-associated methylation sites in islets were in OCRs with different prevalence in islets from T2D cases versus controls. One example is *SLC7A2*, which encodes an amino acid transporter; an OCR ~ 1.5 kb upstream of the transcription start site (TSS) was found only in islets from T2D cases, and DNA methylation was lower in T2D islets (Supplementary Fig. 2a, Supplementary Data 8). Lower DNA methylation is in accordance with a more open chromatin structure, as seen in the T2D islets. In addition, in islets from T2D cases, *WFS1* had more prevalent ATAC-seq peaks, and two sites in this region had lower methylation levels than controls (Supplementary Fig. 2b, Supplementary Data 8). In *PDE7B*, we found eight sites with lower methylation in T2D islets, all located in an OCR covering the TSS (Supplementary Fig. 2c, Supplementary Data 8). We previously found *PDE7B* to be epigenetically upregulated, impairing insulin secretion in T2D[4].

## Most genes differentially expressed in T2D islets also have altered DNA methylation

DNA methylation may interfere with transcriptional activity, as proven by in vitro experiments[4,6] and causal inference test[17], but may also be a marker secondary to altered gene activity[2]. For 97 samples with methylation data in the Islet T2D case-control cohort, we also performed RNA-sequencing (RNA-seq, Fig. 1a). Focusing on protein-coding genes only (*n* = 15,845), we found 203 differentially expressed genes (DEGs) in islets of T2D cases versus controls (*q* < 0.05). We filtered the RNA-seq data to only include DEGs located ±10 kb from the 31,806 sites with differential methylation in the Islet T2D case-control study. This overlap resulted in 550 sites located in or near 153 DEGs; 75% of genes with differential expression in islets from T2D cases versus controls also have one or more sites with differential methylation (Supplementary Data 9). These include genes previously implicated in T2D or islet function, such as *CDKN1C*, *GAD1*, *GLRA1*, *IL6*, *RBP4*, and *SLC2A2*[6,18–21] (Supplementary Fig. 3a–f), suggesting the involvement of DNA methylation in the regulation of genes important for islet function and disease development. Two genes with altered expression in T2D islets had many differentially methylated sites, namely *FOXP1* (30 sites, Fig. 3a) and *SYNPO* (24 sites) (Supplementary Data 9). Notably, 15 of the T2D-associated methylation sites overlap with six OCRs in *FOXP1* (Supplementary Data 8, Fig. 3a). We also found increased methylation of six sites in *TBC1D4* and confirmed lower expression in islets from T2D donors[22] (Supplementary Data 9, Fig. 3b). In addition, *RHOT1* and *CABLES1* had reduced expression together with altered intragenic methylation (Supplementary Data 9, Fig. 3c, d). To visualize a possible link between concurrent differences in methylation and expression, we plotted the relative differences of DNA methylation in the promoter of DEGs in islets of T2D cases versus controls (Supplementary Fig. 3g, Supplementary Data 9). Increased promoter methylation and decreased expression were seen for e.g. *BEST3*, *HHATL* and *SLC2A2*, while e.g. *IGF1*, *SFRP4* and *SYNPO* had reduced promoter methylation and increased expression in islets from T2D donors (Supplementary Fig. 3g).

Next, we calculated a weighted combined methylation risk score (MRS) for each gene with altered expression and more than five differentially methylated CpG sites within or near (±10 kb) that gene, to study the combined effect of multiple differentially methylated sites in individual genes (total 27 genes; Supplementary Data 10). All 27 MRSs were different between T2D cases versus controls in multiple linear regression models ($p < 1.1 \times 10^{-4}$). MRSs for *SYNPO* ($\beta = 84.6 \pm 11.1$, $p = 2.1 \times 10^{-11}$) and *FOXP1* ($\beta = 57.7 \pm 8.1$, $p = 2.2 \times 10^{-10}$) had the biggest effect size in the linear models, displaying that individuals with T2D have a higher MRS compared to controls. Supplementary Fig. 4a–f presents the MRS for *BEST3*, *CABLES1*, *FAIM2*, *FOXP1*, *SLC2A2* and *TBC1D4*, demonstrating the combined effect of several differentially methylated sites on T2D. Moreover, when performing multiple logistic models, higher values of all MRS were associated with a higher risk of having T2D with odds ratios (OR) ranging between 1.03 and 2.66 per 1% increase in methylation ($p < 5.4 \times 10^{-4}$; Supplementary Data 10, Supplementary Fig. 4g).

Focusing on methylation sites associated with both T2D and HbA1c (*n* = 5584), 113 sites were in or near (±10 kb) 65 DEGs in islets from T2D cases versus controls (Supplementary Data 9).

## Differential methylation in regions bound by islet- or β-cell-specific transcription factors

DNA methylation may affect TF binding[23]. To evaluate the possibility of altered methylation in islets from T2D cases affecting TF binding, we

**Table 1 | Clinical characteristics of the pancreatic islet donors included in the Islet T2D case-control cohort and Islet HbA1c cohort**

| Characteristics | Islet T2D case-control cohort | | | Islet HbA1c cohort |
|---|---|---|---|---|
| | Controls | T2D cases | *P*-value | |
| *n* (male/female) | 75 (46/29) | 25 (17/8) | | 114 (73/41) |
| Age (years) | 61.4 [43–81] | 62.8 [45–81] | 0.47 | 58.8 [24–81] |
| BMI (kg/m²) | 25.9 [18–40.1] | 27.4 [21.6–34.9] | 0.11 | 26.3 [18–40.1] |
| HbA1c (mmol/mol) | 36.7 [23–41] | 50.1 [39–86] | $5 \times 10^{-17}$ | 38.8 [23–70] |
| Islet purity (%) | 83.2 [70–100] | 82.6 [70–100] | 0.76 | 83.4 [70–100] |

Data are shown as mean [range]. *P*-values are based on a two-sample *t*-test (two-tailed). Controls have HbA1c < 42 and no diabetes diagnosis.

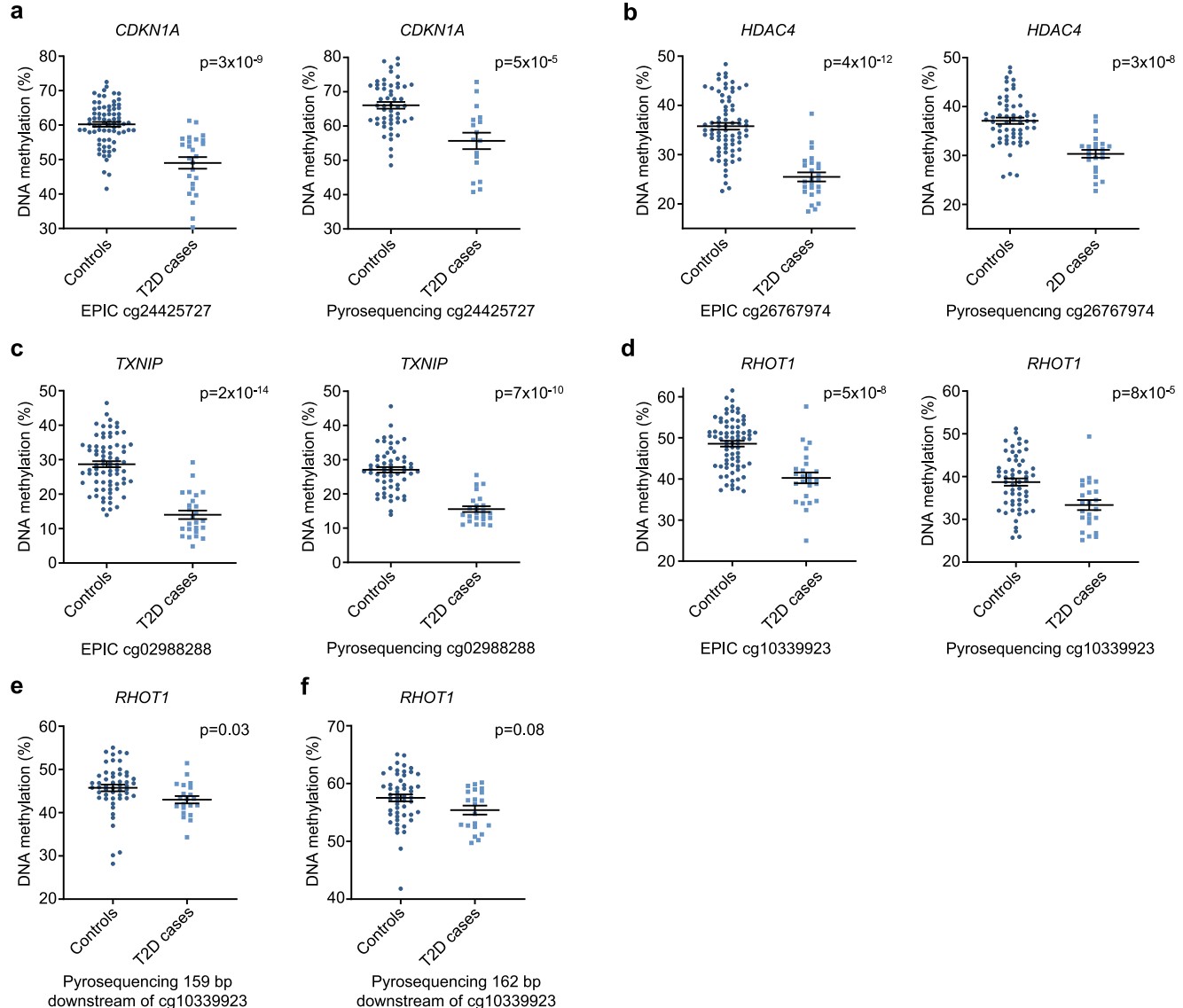

**Fig. 2 | Pyrosequencing validation of EPIC array DNA methylation data.**
cg24425727 of *CDKN1A* (**a**), cg26767974 of *HDAC4* (**b**), cg02988288 of *TXNIP* (**c**), and cg10339923 of *RHOT1* (**d**). **e**, **f** Methylation of sites 159 and 162 bp downstream of cg10339923 in *RHOT1*. Absolute methylation levels (%) and mean ± SEM are given in **a–f**. *P*-values are based on a linear regression, adjusted for age, BMI, sex, days in culture, and islet purity. *n* = 75 controls + 25 T2D cases (EPIC data) or *n* = 59 controls + 24 T2D cases (Pyrosequencing). Source data are provided as a Source Data file.

combined our 31,806 T2D-associated methylation sites with DNA sequences bound by islet-specific TFs (FOXA2, MAFB, NKX2.2, NKX6.1, and PDX1)[24]. We found 3113 T2D-associated methylation sites in the binding regions for any of these TFs (Fig. 4a), with regions bound by each TF overlapping 259–2154 sites (Fig. 4b, c, Supplementary Data 11). Conversely, 4.1–6.4% of regions bound by islet-specific TFs had one or more sites differentially methylated in T2D islets (Fig. 4c). Notably, six T2D-associated methylation sites were found in regions bound by all five islet-specific TFs (Fig. 4b), all with higher methylation in T2D islets, and five were also associated with HbA1c (Supplementary Data 11). Three of these sites are in the same binding region in *LRFN2*, one in *IGF2BP3*, and one in *ASAP1*. Interestingly, *LRFN2* has been linked to T2D and insulin secretion[25].

Next, we evaluated the TF binding regions that overlap T2D-associated methylation sites in genes that are also differentially expressed in T2D islets. We found 4–29 binding regions for respective TFs (Fig. 4c) that overlap 55 methylation sites in 35 DEGs in T2D islets (Supplementary Data 11). Notably, in islets from T2D cases, *BEST3* had 50% lower expression and three sites with increased

methylation in a region near the TSS known to be bound by MAFB, NKX2.2, and PDX1 (Supplementary Fig. 3h). This region resides in an OCR (Supplementary Data 8, Supplementary Fig. 3h). Another example is *FAIM2*, in which the 3'UTR has binding sites for NKX2.2, NKX6.1, and PDX1, which overlap a site with lower methylation in T2D islets. Lower methylation was also found in an intronic MAFB binding region of *FAIM2*, and *FAIM2* expression was increased in T2D islets. Three sites near the TSS of *SLC2A2* (encoding GLUT2) had increased methylation in T2D islets, are in an active enhancer region, and overlap NKX6.1 and PDX1 binding. Furthermore, *SLC2A2* expression was decreased in T2D islets (Supplementary Fig. 3f) and *SLC2A2* is marked by an OCR upstream of the TSS (Supplementary Data 8, Supplementary Fig. 3f). *FOXP1* exhibited lower expression in T2D islets, and three of the T2D-associated methylation sites near the TSS of *FOXP1* are within an OCR with binding sites for NKX2.2 and FOXA2 (Fig. 3a).

These data support a direct regulatory role of T2D-associated epigenetic alterations in TF binding and, thus, expression in human islets.

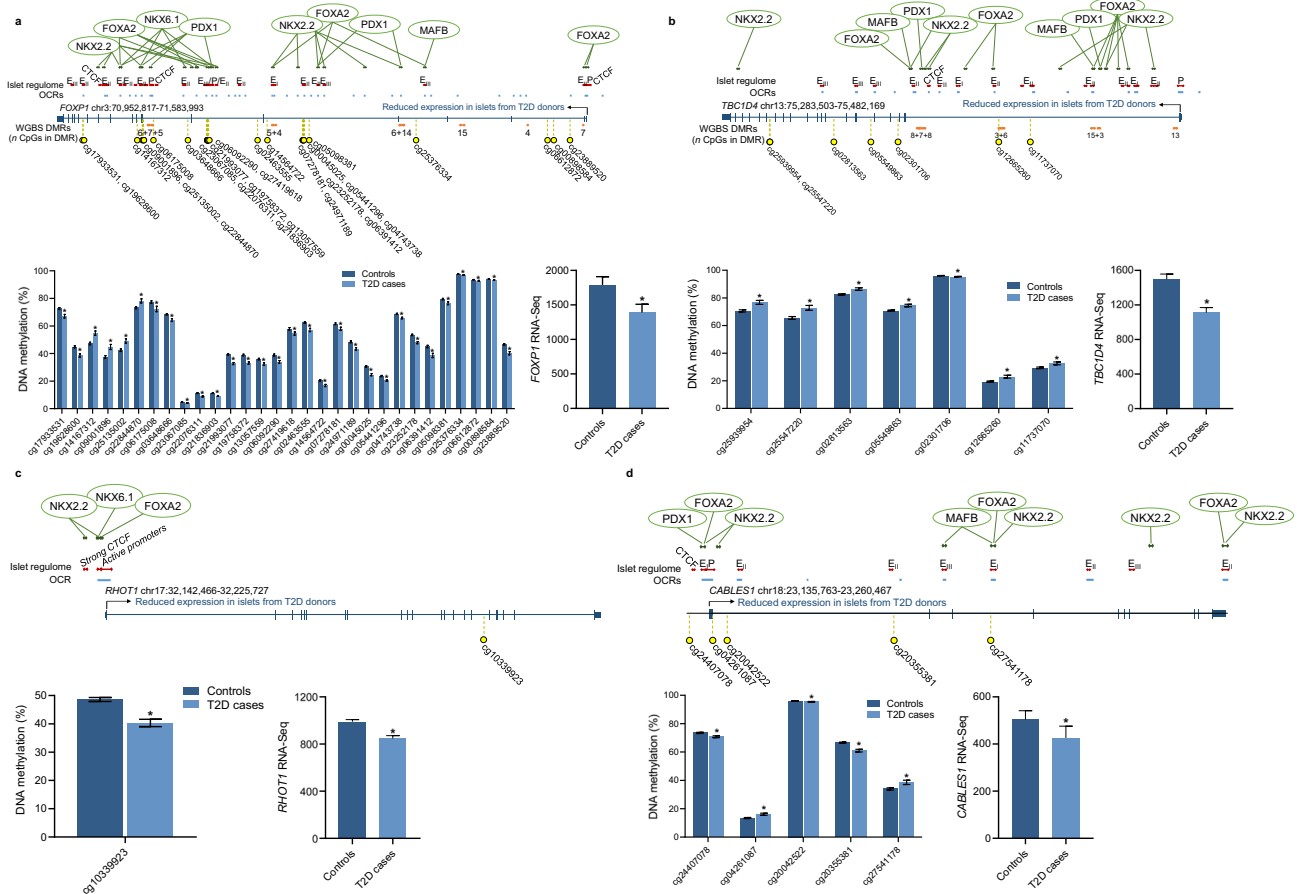

**Fig. 3 | Epigenetic and transcriptional maps of identified T2D candidates.** Differential DNA methylation and expression ($q < 0.05$) in islets from 25 T2D cases vs. 75 controls overlap with islet-specific markers of transcriptional regulation[24, 26], open chromatin regions (OCRs)[16], and differentially methylated regions (DMRs) identified by whole-genome bisulfite sequencing (WGBS) of human islets[6]. **a** *FOXP1* has 30 differentially methylated sites and reduced expression in islets from T2D cases vs. controls. This region overlaps with markers of active enhancers (class I-III; EI-EIII) and promoters (P), OCRs, and five islet-specific transcription factors (TFs). WGBS previously found 10 DMRs in *FOXP1* in islets from T2D cases vs. controls. **b** *TBC1D4* shows increased DNA methylation and decreased expression in islets from T2D cases vs. controls. This gene further displays 20 markers of the islet regulome, eight T2D-associated DMRs, 17 OCRs, and several binding regions for islet-specific TFs. **c** The TSS of *RHOT1* displays an OCR, marks of active promoter and binding sites for three islet-specific TFs. *RHOT1* expression was reduced and an intronic site showed reduced DNA methylation in islets from T2D cases vs. controls. **d** *CABLES1* has altered methylation of five sites and reduced expression in islets from T2D cases vs. controls. TSS and the upstream region are in an OCR and have chromatin marks of active enhancer and promoter, and the region has 11 binding sites for islet-specific TFs. Chromosomal positions are based on hg38. Data presented as mean ± SEM. *$q < 0.05$ for T2D cases ($n = 25$) vs. Controls ($n = 75$ for DNA methylation and 72 for gene expression), based on linear regressions adjusted for age, BMI, sex, days in culture, and islet purity. The *p*-values were corrected for multiple testing with FDR analysis. Source data are provided as a Source Data file.

## Integration of DNA methylation with the islet regulome

As the epigenome comprises several layers of information, DNA methylation should ideally be integrated with other epigenetic marks. Therefore, we combined our methylation data with islet data for H3K27ac, H3K4me1, H3K4me3, Mediator, cohesin, and CTCF binding in OCRs[26] (Fig. 4a, d). Among the T2D-associated methylation sites in our study, 2507 were in regions marked as active promoters, 4121 in active enhancers, and 1305 in strong CTCF regions (Fig. 4d, Supplementary Data 12). H3K27ac and H3K4me3 mark active promoters, whereas Mediator and H3K27ac binding define active enhancer classes I–III[26]. T2D-associated methylation sites of e.g. *BEST3*, *CABLES1*, *CDKN1A*, *CDKN2B*, *DNMT3A*, *FOXP1*, *GLP1R*, *HDAC7*, *INS-IGF2*, *KCNJ11*, *OPRD1*, and *PDX1* overlap active promoters, whereas methylation sites of *HDAC4*, *HDAC7*, *INS-IGF2*, *MEG3*, *PDX1-AS1*, *SLC2A2*, *SYT13*, and *THADA* overlap active enhancers (Supplementary Data 12).

## DNA methylation is associated with future T2D

For DNA methylation to serve as a prognostic marker, it should be analyzed in readily available clinical samples. Therefore, we tested in blood samples whether methylation of 113 sites associated with both

T2D and HbA1c in islets and within 10 kb of DEGs in T2D islets (Supplementary Data 9) are associated with future T2D using a matched case-control study nested within the prospective EPIC-Potsdam cohort study (Supplementary Data 13). We found four sites, annotated to *NKX6.2*, *SYNPO*, *RHOT1*, and *CABLES1*, associated with a lower risk of incident T2D in EPIC-Potsdam (odds ratios <1; $p < 0.05$) and with lower methylation in islets from T2D cases versus controls (Table 2). Islet RNA-seq data showed increased expression of *NKX6.2* and *SYNPO* in T2D islets, where the methylation sites are in promoter regions, and decreased expression of *RHOT1* and *CABLES1* in T2D islets, where the methylation sites are in gene bodies (Supplementary Data 9, Figs. 2d and 3c, d).

We next compared our identified sites showing differential DNA methylation in pancreatic islets from T2D cases versus controls (Supplementary Data 1), with published DNA methylation data from prospective or incident T2D studies performed in blood[27–33]. Previous studies in blood identified two T2D-associated methylation sites (cg00574958/*CPT1* and cg02711608/*SLC1A5*) also identified here in human islets (Supplementary Data 14)[27,31,33]. Moreover, previous T2D studies performed in blood found differential methylation of the same

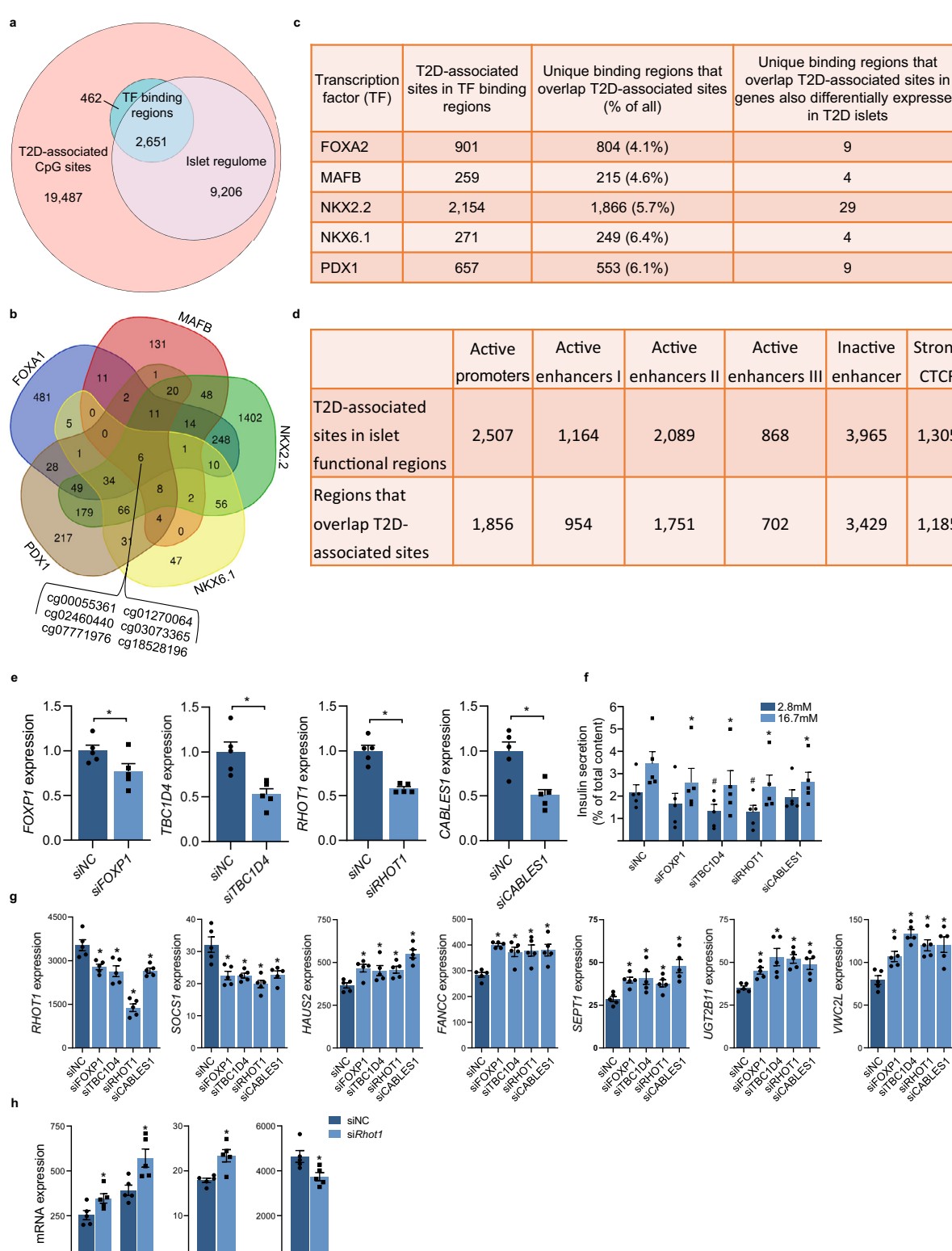

genes as we identified in the Islet T2D case-control cohort (e.g., *TXNIP*, *ABCG1*, and *HDAC4*), but the specific sites identified in blood were different from the methylation sites identified in islets (Supplementary Data 1, 14)[27–33].

### Functional validation of selected candidates in human islets

To explore the mechanisms that contribute to T2D pathogenesis, four genes were selected for functional follow-up based on differential methylation in both the Islet T2D case-control and Islet HbA1c cohorts and altered expression in T2D islets (Supplementary Data 9). We selected *FOXP1*, which has the most differentially methylated sites (*n* = 30; Fig. 3a), *TBC1D4* (Fig. 3b), which has been implicated in T2D[22] but the function of which has not been studied in human islets, and two genes (*RHOT1* and *CABLES1*) that also had sites whose methylation in blood associated with future T2D (Table 2, Figs. 2d and 3c, d). These genes encode a TF (*FOXP1*)[34] and proteins

**Fig. 4 | Differential methylation in the *Islet T2D case-control cohort* in regions with regulatory functions and functional validation of candidate genes in human islets. a** Among methylation sites associated with T2D ($q < 0.05$), 10% are located in islet-specific transcription factor (TF) binding regions ($n = 3113$)[24] and 37% in the islet regulome ($n = 11,857$)[26]. **b** Venn diagram showing T2D-associated methylation sites in regions bound by one or more islet-specific TFs[24]. **c** Regions bound by islet-specific TFs[24] that overlap differentially methylated sites in the Islet T2D case-control cohort ($q < 0.05$). **d** T2D-associated methylation sites from this study ($n = 31,806$) located in regions important for transcriptional regulation in human islets. The islet regulome is based on experimental data in human islets[26]. **e** Expression after silencing *FOXP1*, *TBC1D4*, *RHOT1*, and *CABLES1* in human islets using siRNA. *$p < 0.05$ vs. siNC based on a two-sided paired *t*-test. $p = 0.042$ (si*FOXP1*), $p = 0.002$ (si*TBC1D4*), $p = 0.003$ (si*RHOT1*) and $p = 0.002$ (si*CABLES1*).

**f** Insulin secretion after silencing of *FOXP1*, *TBC1D4*, *RHOT1*, and *CABLES1* in human islets. #$p < 0.05$ vs. siNC at 2.8 mM glucose and *$p < 0.05$ vs. siNC at 16.7 mM glucose based on a two-sided paired *t*-test. $p = 0.011$ (si*FOXP1* at 16.7 mM), $p = 0.025$ and $p = 0.013$ (si*TBC1D4* at 2.8 and 16.7 mM, respectively), $p = 0.016$ and $p = 0.030$ (si*RHOT1* at 2.8 and 16.7 mM, respectively) and $p = 0.013$ (si*CABLES1* at 16.7 mM). **g** Global expression in human islets silenced for *FOXP1*, *TBC1D4*, *RHOT1*, or *CABLES1* using siRNA detected seven genes affected by all four siRNAs. *$p < 0.05$ vs. siNC based on limma moderated *t*-test (paired, two-sided), exact *p*-values in Supplementary Data 15. **h** Silencing of *RHOT1* increased expression of *PINK1*, *TFEB*, and *FIS1*, whereas *OPA1* was downregulated. *$p < 0.05$ based on limma moderated two-sided paired *t*-test (exact *p*-values in Supplementary Data 15). Data in **e**–**h** are shown as mean ± SEM, $n = 5$ independent experiments in different islet donors. Source data are provided as a Source Data file. siNC non-target control.

**Table 2 | DNA methylation associated with T2D in the Islet T2D case-control cohort within genes differentially expressed in islets from T2D cases vs. controls that also associated with future T2D based on blood DNA methylation in the prospective EPIC-Potsdam T2D case-control study**

| CpG site | Islet T2D case-control cohort | | | | | | EPIC-Potsdam | |
| | DNA methylation | | | | RNA-seq | | Blood DNA methylation | |
| | Absolute difference | *P*-value | Odds ratio (95% CI) | *P*-value | Gene | *P*-value | Odds ratio (95% CI) | *P*-value |
|---|---|---|---|---|---|---|---|---|
| cg05324789 | −3.50% | $3.2 \times 10^{-4}$ | 0.77 (0.65–0.90) | 0.0014 | *NKX6.2* | $4.3 \times 10^{-5}$ | 0.43 (0.24–0.76) | 0.004 |
| cg20355381 | −5.60% | $4.4 \times 10^{-7}$ | 0.63 (0.47–0.78) | $3.1 \times 10^{-4}$ | *CABLES1* | $1.2 \times 10^{-4}$ | 0.56 (0.37–0.84) | 0.006 |
| cg15167202 | −4.70% | $6.8 \times 10^{-4}$ | 0.85 (0.76–0.94) | 0.0027 | *SYNPO* | $3.0 \times 10^{-4}$ | 0.43 (0.20–0.94) | 0.034 |
| cg10339923 | −8.30% | $5.0 \times 10^{-8}$ | 0.76 (0.66–0.86) | $4.9 \times 10^{-5}$ | *RHOT1* | $3.3 \times 10^{-4}$ | 0.68 (0.47–0.97) | 0.035 |

For islet DNA methylation, both absolute difference (cases-controls) based on linear regression and odds ratios based on logistic regression are shown.

that regulate glucose homeostasis (*TBC1D4*)[22], affect mitophagy and mitochondrial function (*RHOT1*)[35], and impact the cell cycle (*CABLES1*)[36].

To mimic the situation in T2D, *FOXP1*, *TBC1D4*, *RHOT1*, and *CABLES1* were silenced in human islets using siRNA (Fig. 4e), resulting in reduced glucose-stimulated insulin secretion (GSIS) (Fig. 4f) without changing the insulin content (Supplementary Fig. 5a). To further identify cellular functions affected by silencing of these genes, we compared global expression in islets silenced for each gene versus control (Supplementary Data 15, Supplementary Fig. 5b). Interestingly, *RHOT1* and *SOCS1* expression was decreased by all four siRNA in human islets, whereas expression of *HAUS2*, *FANCC*, *SEPT1*, *UGT2B11*, and *VWC2L* was increased in all siRNA experiments (Fig. 4g). These data support *RHOT1* as an important factor in human islet function.

Moreover, *RHOT1* silencing in human islets upregulated *PINK1*, *TFEB*, and *FIS1*, genes that, together with *RHOT1*, are part of the mitophagy pathway (Fig. 4h)[37]. FIS1 is also involved in mitochondrial fission, whereas *OPA1*, the protein product of which affects mitochondrial fusion, was downregulated in *RHOT1*-deficient islets (Fig. 4h). Of importance for mitochondrial function, *RHOT1*-deficient islets had reduced expression of genes in the electron transport chain (e.g., *COX7B*, *NDUFA1*, *NDUFB3*, and *NDUFC2*; Supplementary Data 15). Moreover, genes encoding epigenetic enzymes (e.g., *DNMT3A* and *HDAC1*) and regulators of insulin secretion (e.g., *CREB5* and *PIK3CB*) were differentially expressed in *RHOT1*-deficient islets (Supplementary Data 15).

To follow up on the relationship between *RHOT1* and mitochondrial dynamics, we correlated expression of *RHOT1* with genes involved in mitophagy and mitochondrial fusion and fission[37], using RNA-seq data from the Islet HbA1c cohort. Here, expression of *LAMP2*, a lysosome protein, and *MFN2*, a mitochondrial fusion protein, correlated positively with *RHOT1* expression (Supplementary Fig. 5c, d).

*FOXP1* silencing in islets affected T2D-related genes, e.g. downregulated *HK1* and *PRKCD* and upregulated *MAPK8* and *DNMT3A* ($p < 0.05$; Supplementary Data 15). *TBC1D4* silencing in islets downregulated *HK1* and upregulated genes encoding metallothioneins (e.g., *MT1A*, *MT1B*, *MT1E*, and *MT1G*), which regulate zinc homeostasis and protect from reactive oxygen species[38] (Supplementary Data 15). In *CABLES1*-deficient islets, two of the genes which were regulated individually by all four siRNAs were among the top deregulated genes, *HAUS2* and *SEPT1* (Fig. 4g, Supplementary Data 15).

To characterize potential mechanisms underlying the reduced *RHOT1* expression when silencing *FOXP1*, *TBC1D4*, and *CABLES1* in human islets, we identified 619 TF binding sites in the vicinity of the *RHOT1* gene (−1000, 0) using publicly available ChIP-seq data from the Gene Transcription Regulation Database (GTRD)[39] (Supplementary Fig. 5e). Since most experiments in the GTRD were performed in non-islet cell types, we examined whether the investigated genomic regions are accessible in human islets[26]. We identified 362 TFs with 614 binding sites within regions defined as promoters in pancreatic islets (Supplementary Data 16). Next, we searched for overlaps between the 362 TFs that could bind 0–1000 bp upstream of *RHOT1* TSS and differentially expressed genes after silencing *FOXP1*, *CABLES1*, or *TBC1D4* in human islets (Supplementary Data 15, 16). We detected 7, 8 and 13 differentially expressed TFs, and among these, 6, 4 and 8 were downregulated after silencing *FOXP1*, *CABLES1* or *TBC1D4*, respectively (Supplementary Data 16, Supplementary Fig. 5e). Although dysregulation of each of these TFs could contribute to the reduction of *RHOT1*, one TF, MLXIP (formerly known as MONDOA), was found to be dysregulated in all 3 silencing experiments. MLXIP is an important glucose-sensing TF in human β-cells[40].

Together, these data show that mimicking the expression changes identified in islets from individuals with T2D results in impaired insulin secretion and altered expression of key genes in human islets.

## Characterization of Rhot1-deficient INS-1 832/13 β-cells

Our human islet experiments, where *RHOT1* expression and insulin secretion were reduced by silencing all four selected candidates, together with altered methylation and expression in T2D islets, suggest that RHOT1 is a key regulator of insulin secretion in T2D. Therefore, we thoroughly dissected the role of RHOT1 deficiency in β-cells. *RHOT1* encodes a Rho GTPase, regulating calcium-dependent mitochondrial transport and mitochondrial quality control (QC) and dynamics[35]. First, we investigated the direct impact of increased promoter DNA methylation on the transcriptional activity of *RHOT1*, using a luciferase assay (Supplementary Methods). The CpG-free luciferase reporter construct including 1540 bp of the *RHOT1* promoter was methylated by SssI and transfected into INS-1 832/13 β-cells. Methylation of the promoter reduced the transcriptional activity (Supplementary Fig. 5f), supporting that DNA methylation can influence gene expression. Of note, the promoter sites methylated in the Luciferase experiment are different from the intronic *RHOT1* methylation site associated with T2D in human islets. CpG sites of different gene regulatory regions, although within the same gene, have different ways of influencing gene expression. For example, elevated promoter methylation has been shown to reduce expression, while intragenic methylation has been linked to higher expression[2,6]. Due to low Epic array coverage directly upstream of the *RHOT1* TSS, we cannot conclude if T2D is associated with differential *RHOT1* promoter methylation or not.

We then silenced *Rhot1* in INS-1 832/13 β-cells (hereafter called *Rhot1*-deficient β-cells) and monitored insulin secretion and cell function. GSIS was perturbed in *Rhot1*-deficient β-cells when normalizing to total protein or insulin content (Fig. 5a, b, Supplementary Fig. 5g). *Rhot1* silencing was also verified by Western blot showing 73% reduction in the protein (Fig. 5c, Supplementary Fig. 5h, i). Stimulating β-cells with K$^+$ had no impact on insulin secretion in *Rhot1*-deficient β-cells relative to control (Fig. 5d), indicating that the defect in *Rhot1*-deficient β-cells lies proximal to closing of the K$_{ATP}$ channels, potentially in glucose metabolism or mitochondrial function.

Thus, we focused on mitochondria. High-content automated microscopy using MitoTracker Deep Red staining found a reduced mitochondrial count and area in *Rhot1*-deficient β-cells (Fig. 5e). Accordingly, the protein levels of citrate synthase, a mitochondrial matrix TCA cycle enzyme and marker of intact mitochondria, were reduced in *Rhot1*-deficient β-cells (Supplementary Fig. 6a). *Rhot1* silencing also reduced protein levels of cytochrome C and components of the electron transport chain (Supplementary Fig. 6b, c). These findings may explain the impaired glucose-stimulated ATP production in *Rhot1*-deficient β-cells determined by PercevalHR tracing (Fig. 5f). In addition, the mitochondrial Ca$^{2+}$ response to high glucose was lower in *Rhot1*-deficient β-cells (Fig. 5g). Using Seahorse, we found reductions in the oxygen consumption rate, basal mitochondrial respiration, and ATP-coupled respiration in *Rhot1*-deficient β-cells (Fig. 5h). Calcium uptake into the mitochondria is ATP-independent. However, when ATP levels are low, cytosolic Ca$^{2+}$ dynamics are impacted, leading to reduced mitochondrial calcium uptake. To test this, we measured cytosolic Ca$^{2+}$, finding a lower cellular response to glucose after *Rhot1* silencing (Fig. 5i). As mitochondrial calcium uptake was lower also upon addition of K$^+$ (Fig. 5g), it is likely affected directly by *Rhot1* silencing. MitoSox Red staining followed by flow cytometry detected increased mitochondrial superoxide production in *Rhot1*-deficient β-cells ($p < 0.005$; Supplementary Fig. 6d–f).

As RHOT1 has been implicated in mitophagy[41], we analyzed key autophagy markers Lamp1, Lc3b, p62, Parkin, and Pink in *Rhot1*-deficient and control β-cells, both under normal culture conditions and after 3 h of amino acid starvation (inducing autophagy). Protein levels of lysosomal marker Lamp1 increased under both normal and starved conditions in *Rhot1*-deficient β-cells (Supplementary Fig. 7a). Starvation led to a 50% decrease in p62 protein in control β-cells, as expected,

but this was impaired after *Rhot1* silencing, causing higher p62 levels than in control cells after starvation (Supplementary Fig. 7b), indicating impaired autophagic activity in *Rhot1*-deficient cells. As expected, starvation increased Lc3b-II, a smaller lipidated isoform, in control β-cells, and a similar result occurred when silencing *Rhot1* (Supplementary Fig. 7c). There were no differences in Parkin (relative expression, mean ± SEM: siNC 1.00 ± 0.11, siRhot1 1.10 ± 0.07, $p = 0.26$ with a two-tailed paired *t*-test, $n = 12$) or Pink1 (relative expression, mean ± SEM: siNC 1.00 ± 0.05, siRhot1 0.92 ± 0.07, $p = 0.11$ with a two-tailed pared *t*-test, $n = 7$) levels (analyzed under normal conditions only). Considering the altered expression of genes involved in mitochondrial fission/fusion in *RHOT1*-deficient human islets (Fig. 4h) and changes in mitochondrial number in *Rhot1*-deficient β-cells (Fig. 5e), we analyzed Opa1 protein levels. We found a slight increase in the shorter isoform and a lower ratio of the levels of the longer and shorter isoforms of Opa1 in *Rhot1*-deficient β-cells (Supplementary Fig. 7d). Both isoforms are needed for Opa1 function, and cleavage of the longer isoform to produce the shorter isoform is triggered by loss of mitochondrial membrane potential[42].

Collectively, these data support a model in which lower RHOT1 levels in β-cells lead to deregulation of mitophagy and altered expression of proteins involved in mitochondrial dynamics and metabolism, resulting in mitochondrial dysfunction and perturbed GSIS.

## Metabolomics in Rhot1-deficient β-cells

Next, we studied metabolomic disturbances occurring in *Rhot1*-deficient β-cells. We generated GC-MS and LC-MS data for 248 unique metabolic features (Supplementary Data 17). A total of 53 and 37 metabolites were increased and 32 and 54 decreased in *Rhot1*-deficient versus control β-cells at 2.8 mM and 16.7 mM glucose, respectively. In addition, 40 metabolites differed in their response to glucose between *Rhot1*-deficient versus control β-cells, with 34 metabolites showing a reduced response (Supplementary Data 17). Figure 6a shows overlap between metabolites affected by *Rhot1* silencing at 2.8 mM, 16.7 mM, and response to high glucose.

One key finding was lower basal levels of L-proline and no response of proline levels to high glucose in *Rhot1*-deficient β-cells (Fig. 6b, Supplementary Data 17). Proline is an amino acid derived from glutamate in an ATP-dependent process, which can activate glycine and glutamate receptors. As glutamic acid was unchanged in *Rhot1*-deficient β-cells (Supplementary Data 17), the reduced ATP production observed in these cells may cause proline deficiency. Other affected metabolites include glycine, gamma-aminobutyric acid (GABA), and myo-inositol, which were all lower in *Rhot1*-deficient β-cells (Fig. 6c-e, Supplementary Data 17). Notably, a glycine receptor, *GLRA1*, had higher DNA methylation and lower expression in T2D islets in this study (Supplementary Fig. 3c, Supplementary Data 9) and when human islets were exposed to glucotoxic conditions[18]. *Glra1* silencing decreased and glycine exposure increased GSIS[18], supporting a need for glycine in achieving appropriate insulin secretion. GABA, a neurotransmitter produced in β-cells, impacts β-cell survival and regeneration[43]. In this study, a GABA receptor gene, *GABRA2*, had altered methylation and lower expression in T2D islets (Supplementary Data 9). Our pathway analysis of genes differentially methylated in T2D islets highlighted purine metabolism (Fig. 1f, Supplementary Data 2), which is in line with the metabolomics analysis of *Rhot1*-deficient β-cells showing altered levels of guanosine and inosine monophosphate (Supplementary Data 17).

Another key finding was upregulation of 24 unique carnitines, including L-carnitine and deoxycarnitine, in *Rhot1*-deficient β-cells (Fig. 6f). Upregulated carnitines indicate active beta-oxidation, i.e., using fat in addition to glucose as a source of energy; the balance tips towards fat utilization when there is a lack of ATP. These findings

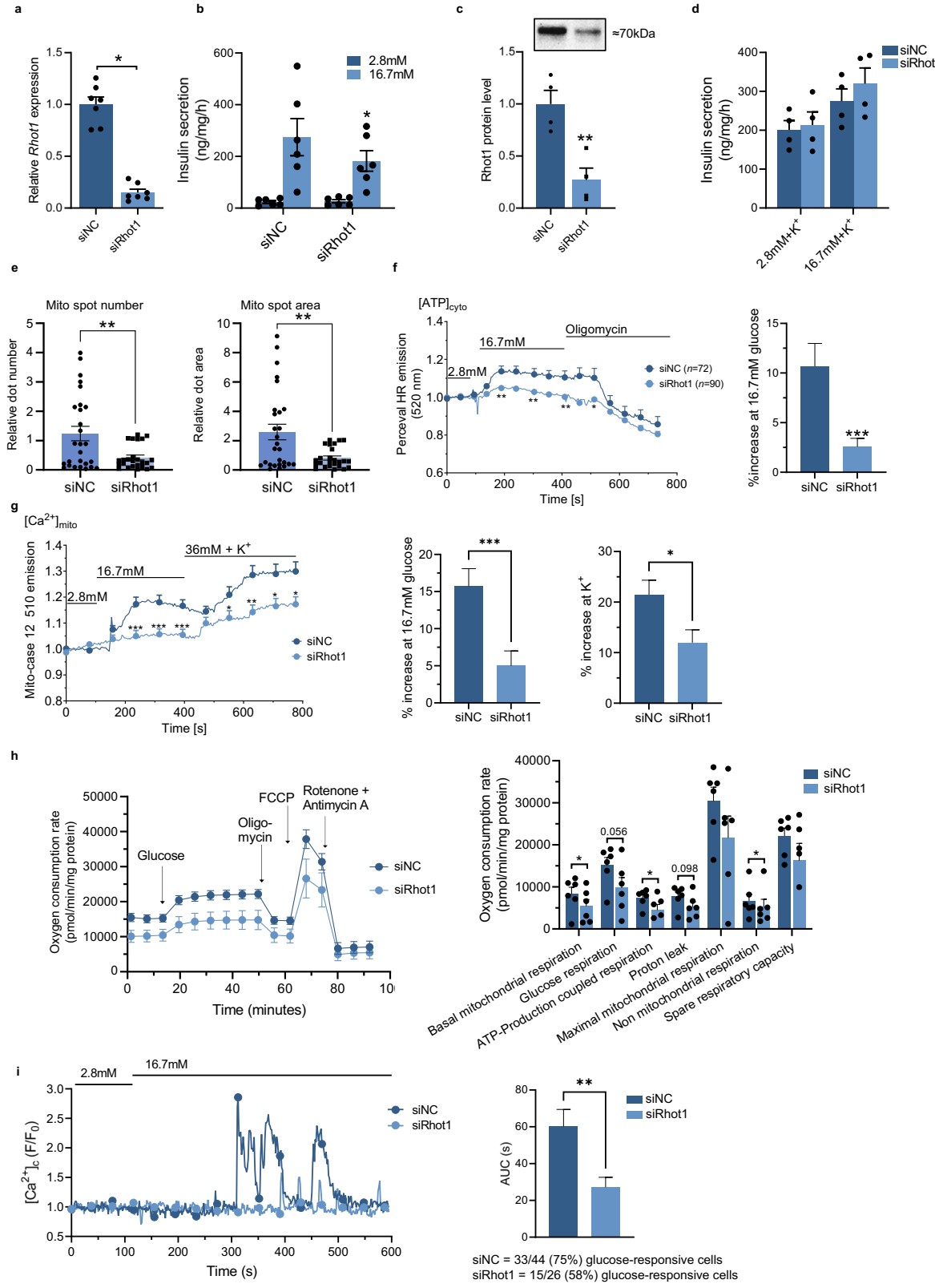

support that *Rhot1*-deficiency impaired glucose metabolism and mitochondrial function.

## Rodent model of diabetes confirms Rhot1-deficiency in islets

To explore RHOT1 in vivo, we studied the Goto-Kakizaki (GK) rat, a model of non-insulin-dependent diabetes[44]. We found an almost complete ablation of Rhot1 protein levels in islets of adult GK rats,

whereas young GK rats, just developing diabetes, had intermediate levels (Fig. 6g, Supplementary Fig. 7e). GK rats also had increased blood glucose levels and lower body weight (Fig. 6h, i); high glucose levels were due to reduced insulin secretion (Fig. 6j). Furthermore, insulin content was reduced by 50% already in young GK rats and deteriorated with age (Fig. 6k). The Rhot1 protein levels in islets correlated positively with GSIS in Wistar rats ($r = 0.90$ and $p = 0.036$), but

**Fig. 5 | Functional studies of *Rhot1*-deficient β-cells. a** Silencing *Rhot1* using siRNA in INS-1 832/13 β-cells reduced *Rhot1* expression. *n* = 7 experiments with one biological replicate for each condition. \**p* < 0.05 (*p* = 0.016) based on a two-sided Wilcoxon matched-pairs signed rank test. **b** Insulin secretion normalized to total protein content in *Rhot1*-deficient β-cells. *n* = 6 experiments with three biological replicates for each condition. \**p* < 0.05 (*p* = 0.031) for siRhot1 vs. siNC at 16.7 mM glucose based on a two-sided Wilcoxon matched-pairs signed rank test. **c** RHOT1 protein levels in *Rhot1*-deficient β-cells with the target signal normalized to the total amount of protein in each lane. A representative Western blot is shown. \*\**p* = 0.005 for siRhot1 vs. siNC based on a two-sided *t*-test, *n* = 4 experiments with one biological replicate. **d** K⁺-stimulated insulin secretion in *Rhot1*-deficient β-cells, *n* = 4 experiments with three biological replicates. **e** MitoTracker staining revealed reduced mitochondrial number (spot number) and size (spot area) in *Rhot1*-deficient β-cells. \*\**p* < 0.005 for siRhot1 vs. siNC based on a two-sided Mann-Whitney test, *n* = 3 passages with 3 × 3 wells each. *p* = 0.0035 (spot number) and *p* = 0.0028 (spot count). **f** Reduced ATP production in *Rhot1*-deficient β-cells, \*\*\**p* < 0.001 (*p* = 0.0006), and **g** reduced mitochondrial Ca²⁺, \*\*\**p* < 0.001 (*p* = 0.0010) and \**p* < 0.05 (*p* = 0.0185) based on a two-sided *t*-test, *n* = 110 (siNC) + 91 (siRhot1) cells from 4 different experiments. **h** Reduced oxygen consumption rate in *Rhot1*-deficient β-cells. \**p* < 0.05 based on a two-sided paired *t*-test, *n* = 6 experiments with 3-4 biological replicates. *p* = 0.0216 (basal respiration), *p* = 0.0414 (ATP-coupled respiration) and *p* = 0.0215 (non-mitochondrial respiration). **i** Cytosolic Ca²⁺ measurements, *n* = 44 (siNC) and *n* = 26 (siRhot1). \*\**p* < 0.005 (*p* = 0.0046) based on a two-sided Mann-Whitney *U* test. AUC area under the curve. Source data are provided as a Source Data file. siNC=non-target control. Data shown as mean ± SEM throughout.

the correlation was insignificant in adult GK rats (*r* = 0.35, *p* = 0.65), in which both Rhot1 and insulin secretion levels are very low. Moreover, the protein level of citrate synthase, a mitochondrial matrix TCA cycle enzyme and marker of intact mitochondria, was significantly reduced in islets from adult GK versus Wistar rats, while the reduction in adult versus young GK rats was nominal (Fig. 6l, Supplementary Fig. 7f). We also found reduced protein levels of components from all five complexes in the electron transport chain in islets from adult GK versus Wistar rats, of which four were also reduced in adult versus young GK rats (Fig. 6m–q, Supplementary Fig. 7g). These results are in line with the data in *Rhot1*-deficient β-cells and support mitochondrial dysfunction in islets from the GK rat. This rodent model mimics human T2D and supports the importance of RHOT1 in islets.

## Discussion

Based on extensive epigenetic analyses in islets from a T2D case-control cohort and a cohort of individuals not previously diagnosed with T2D but with differences in HbA1c levels, we identified novel DNA methylation alterations that can impact T2D progression. By adding additional levels of epigenetic marks and transcriptional activity, subsequently dissecting cellular function and metabolism in human islets, clonal β-cells, and a diabetic rat model, we discovered that *RHOT1* has a key role in islet function and T2D pathogenesis (Fig. 7). DNA methylation of *RHOT1* in blood also associated with future T2D.

This study is the first to show that both T2D and HbA1c, a predictor of diabetes[45], impact DNA methylation of the same sites in human islets. The identified methylation sites are enriched in central pathways, such as Calcium signaling, Type II diabetes mellitus, and Pancreatic secretion, supporting their role in T2D development. By combining our methylation data with RNA-seq and ATAC-Seq data[16] from the same islet cohort, and chromatin marks and TF binding data from another islet source[24,26], we found regions of interest for disease pathogenesis. Overall, we show how altered methylation in islets from T2D cases is found in active promoters and enhancers.

The transcriptional activity of a gene region is the result of several regulatory features, including the presence of DNA methylation, enhancers, and promoters, which determine open or closed chromatin. Altered DNA methylation in regions of T2D candidates bound by islet-specific TFs, as seen for *LRFN2* in this study, with increased methylation in a region bound by all studied TFs (FOXA2, MAFB, NKX2.2, NKX6.1, and PDX1), can be a mechanism leading to reduced islet expression in T2D and, thus, impaired insulin secretion[25]. We also found that 75% of genes with differential expression in islets from T2D cases have altered DNA methylation, suggesting that these genes are epigenetically regulated. Some of these genes are implicated in T2D, including *IGF1*, *GLRA1*, *RBP4*, and *SLC2A2*[18,46,47]. For *SLC2A2*, which encodes GLUT2, we found decreased expression and increased methylation of three sites in an enhancer region overlapping NKX6.1 and PDX1 binding sites in T2D. Mutations in *SLC2A2* cause neonatal diabetes[48]. It should be noted that the effect of DNA methylation depends on the exact location of the

methylated site[2,6]. For example, promoter methylation is generally repressive while gene body methylation has been associated with increased gene expression. Two adjacent CpG sites can also have opposite effects, should they for example be located in the binding sites of a transcriptional activator or repressor, respectively.

Our study is also strengthened by the addition of new epigenetic data to previous findings on genes important for insulin secretion and T2D. For example, we discovered increased methylation of six sites near *TBC1D4* and could confirm lower *TBC1D4* expression in T2D islets[22]. We also discovered reduced GSIS in human islets after *TBC1D4* silencing. Accordingly, GSIS was reduced in *TBC1D4*-deficient EndoC-βH1 cells[49]. Moreover, a previous study found 18 methylation sites in blood associated with future T2D[27]. Of these, two were among our T2D-associated methylation sites in human islets (cg00574958/*CPT1A* and cg02711608/*SLC1A5*), and the first one was also associated with HbA1c. Two additional studies identified T2D-associated methylation of cg00574958/*CPT1A* in blood[31,33]. Other studies performed in blood identified methylation sites associated with T2D, and several of these sites were annotated to the same genes, but for different CpG sites, as were differentially methylated in human islets of T2D donors in the present study[27–33]. Epigenetic alterations common for several tissues may provide therapeutic or predictive biomarkers.

Notably, this study identified numerous novel epigenetic markers associated with both T2D and HbA1c and could technically validate lower methylation of *CDKN1A*, *HDAC4*, *TXNIP*, and *RHOT1* in T2D islets. TXNIP is linked to oxidative stress; *TXNIP* methylation in blood is associated with future T2D[27], and we found lower *TXNIP* methylation in the severe insulin-deficient diabetes subgroup of T2D[50]. We also replicated data from our smaller islet case-control cohorts in this larger cohort, including differential methylation of *CDKN1A*, *GLP1R*, *HDAC7*, *KCNQ1*, *PDE7B*, and *PDX1* in T2D islets, supporting the robustness of our data[4,6]. Moreover, mimicking T2D-associated alterations in *CDKN1A*, *HDAC7*, and *PDE7B* in β-cells resulted in impaired insulin secretion[4,51].

In the present study, we also included the results after adjusting for cell composition using a reference-free method and taking into account unknown sources of technical variation[11]. However, our previous study found no difference in β-cell content between pancreatic islets from donors with T2D versus non-diabetic controls[4].

After combining several data sets and performing functional validation in human islets, we focused on unravelling the mechanisms of RHOT1 and its contribution to impaired GSIS and T2D. Interestingly, *RHOT1* expression and GSIS were reduced by silencing all four selected T2D candidates (*FOXP1*, *TBC1D4*, *RHOT1*, and *CABLES1*) in human islets, supporting RHOT1 as a key player in insulin secretion. The reduction in *RHOT1* may have different causes; for example, FOXP1 is a TF with a binding site in the *RHOT1* promoter, whereas TBC1D4 and RHOT1 display a protein-protein interaction[52]. To find more support for the role of RHOT1 in insulin secretion and T2D, we silenced *Rhot1* in β-cells and evaluated a rodent model of diabetes.

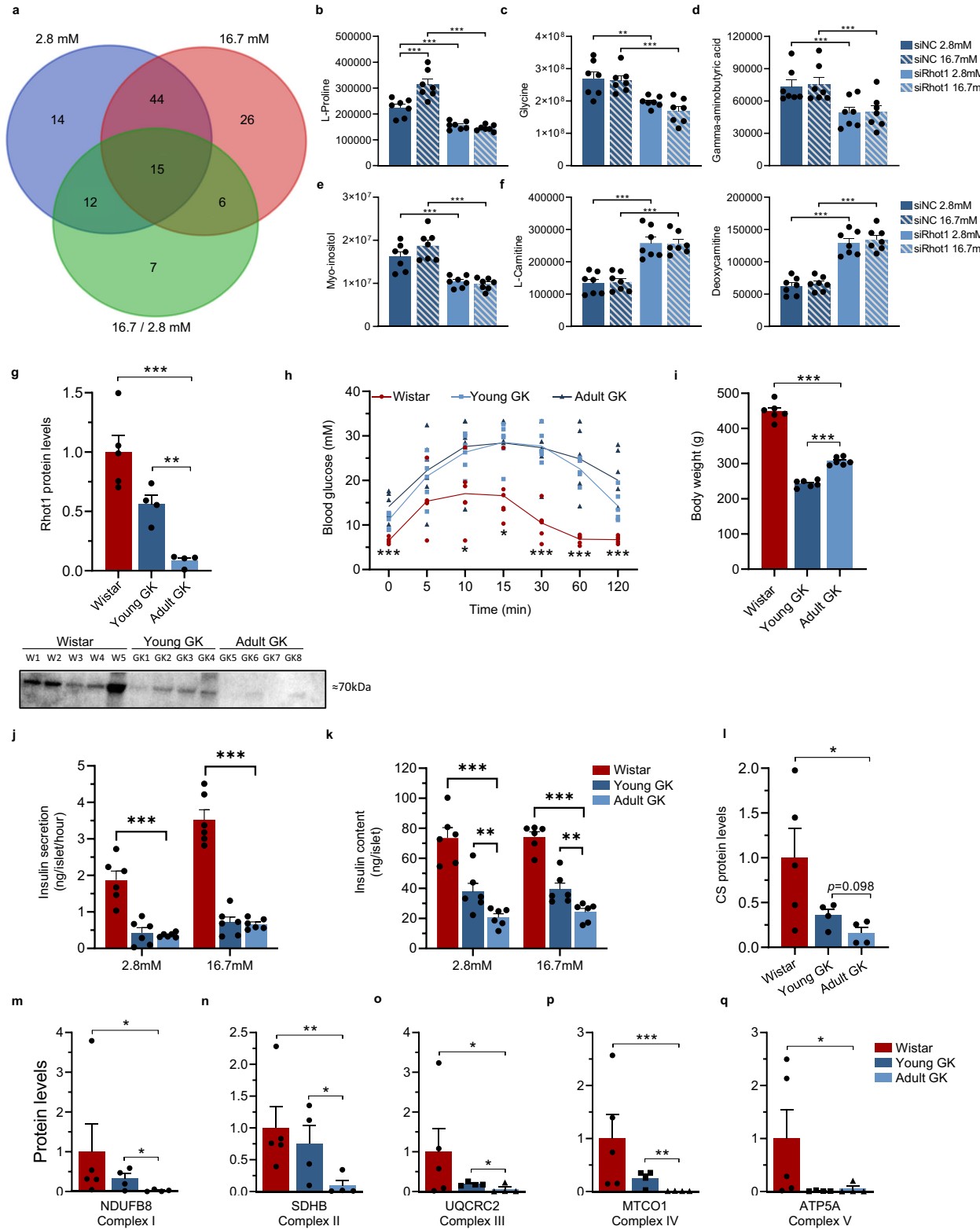

Several aspects of mitochondrial function were impaired in *Rhot1*-deficient β-cells: mitochondrial number and size, ATP production, and expression of proteins in the electron transport chain. The mitochondrial network is regulated by a dynamic process in which the balance of fission and fusion maintains healthy mitochondria while degrading non-functioning organelles through mitophagy[53]. Mitophagy also protects against inflammatory injury and mitochondrial damage, thereby promoting β-cell survival[54]. RHOT1 is an outer

mitochondrial membrane protein important for movement along the cytoskeleton[35]. In *RHOT1*-deficient islets, mitochondrial fission 1 protein (*FIS1*) was upregulated and *OPA1*, encoding a fusion protein, downregulated. More fission than fusion is a sign of cellular stress; mitochondria become smaller and less able to provide ATP[53]. This was further supported by increased ROS production in our *Rhot1*-deficient β-cells. We also found an age-related decline in Rhot1 and lower levels of several components of the electron transport chain in islets from

**Fig. 6 | *Rhot1*-deficient β-cells and a rodent T2D model show that *Rhot1*-deficiency results in metabolic disturbances. a** Summary of metabolomic changes in *Rhot1*-deficient INS-1 832/13 β-cells. *Rhot1*-deficient β-cells have altered L-proline (**b**), glycine (**c**), gamma-aminobutyric acid (GABA) (**d**), myo-inositol (**e**), and carnitine levels (**f**). **$p < 0.005$ and ***$p < 0.001$ for siRhot1 vs. siNC at the indicated glucose level (2.8 or 16.7 mM) based on a two-sided paired *t*-test, $n = 7$. Exact *p*-values are presented in Supplementary Data 17. siNC non-target control. **g** Rhot1 protein in 8- and 12-week-old (young and adult, respectively; $n = 4 + 4$) GK rats and 12-week-old control Wistar rats ($n = 5$) with the target signal normalized to the total amount of protein in each lane. A representative Western blot is shown. **$p < 0.01$ and ***$p < 0.001$ based on two-sided *t*-tests, $p = 0.0008$ (Wistar vs. adult GK) and $p = 0.0011$ (young vs. adult GK). **h, i** Metabolic phenotyping found higher glucose levels during intravenous glucose tolerance in adult GK vs. Wistar rats ($p = 0.00012$, $p = 0.028$, $p = 0.013$, $p = 0.00020$, $p = 0.00011$ and $p = 0.00067$ at 0, 10, 15 30, 60 and 120 min, respectively) (**h**) and lower body weight ($p = 2.3 \times 10^{-7}$ in adult GK vs. Wistar and $p = 1.7 \times 10^{-6}$ in adult vs. young GK) (**i**). Insulin secretion (**j**) and content (**k**) in islets of GK rats was reduced at basal (2.8 mM; $p = 0.00011$ for secretion and $p = 2.7 \times 10^{-5}$ for content) and high (16.7 mM; $p = 1.7 \times 10^{-6}$ for secretion and

$p = 3.7 \times 10^{-7}$ for content) glucose levels. Insulin content declined with age ($p = 0.019$ at 2.8 mM and $p = 0.010$ at 16.7 mM for young vs. adult GK). *$p < 0.05$, **$p < 0.01$, ***$p < 0.001$ based on two-sided *t*-tests, $n = 6$ in each group for data in **h–k**. **l** Citrate synthase protein levels were lower in islets from adult GK ($n = 4$) vs. Wistar ($n = 5$) male rats ($p = 0.02$) and there was a nominal reduction in adult ($n = 4$) vs. young GK ($n = 4$) male GK rats ($p = 0.098$) based on two-sided *t*-tests after log-transformation to reach normal distribution. Target signal was normalized to the total amount of protein loaded in each lane. **m–q** Reduced protein levels of components of the five electron transport chain complexes in islets from adult GK ($n = 4$) vs. control Wistar rats ($n = 5$) as well as in adult vs. young ($n = 4$) GK rats, based on two-sided *t*-tests after log-transformation to reach normal distribution. *$p < 0.05$, **$p < 0.005$, and ***$p < 0.001$. Target signal was normalized to the total amount of protein loaded in each lane. For adult GK vs. Wistar, $p = 0.012$ (NDUFB8), $p = 0.0043$ (SDHB), $p = 0.049$ (UQCRC2), $p = 0.0003$ (MTCO1) and $p = 0.045$ (ATP5A). For adult vs. young GK, $p = 0.043$ (NDUFB8), $p = 0.031$ (SDHB), $p = 0.033$ (UQCRC2) and $p = 0.0022$ (MTCO1). Data shown as mean ± SEM throughout. Source data are provided as a Source Data file.

diabetic GK rats, supporting the role of Rhot1 in islet function and T2D. Yet another rodent model supported our study[41]. Studies in neurons have shown that RHOT1 connects the mitochondria via motor proteins to the axon filaments, thereby regulating mitochondrial movement[55]. Therefore, it is likely that RHOT1 is needed for proper mitochondrial movement also in human islets and β-cells, creating a balance in mitochondrial dynamics and maintaining functional mitochondria, as required to maintain the proper ATP production needed for GSIS. Thus, impaired *RHOT1* expression alters mitochondrial dynamics, which impacts the number of functional mitochondria, resulting in reduced ATP production and insulin secretion (Fig. 7d).

We also studied the metabolome of *Rhot1*-deficient β-cells. A striking finding was upregulation of acylcarnitines, which are linked to beta-oxidation, connecting Rhot1-deficiency in β-cells to mitochondrial metabolism. Other key findings were lower L-proline, GABA, and glycine levels in *Rhot1*-deficient β-cells. In line with this, glycine levels are lower in T2D patients[56] and, in our study, *GLR1A*, which encodes a glycine receptor, was downregulated and had increased methylation in T2D islets. Notably, glycine and n-acetylcysteine supplementation improved mitochondrial function in T2D patients[57]. GABA, a neurotransmitter also produced in β-cells, regulates islet cell function[43]. We found lower GABA levels in *Rhot1*-deficient β-cells, together with differential methylation and reduced expression of *GABRA2*, a GABA receptor, in T2D islets. Our data support a model in which RHOT1-deficiency impairs insulin secretion via mitochondrial dysfunction and numerous metabolic alterations.

While GK rats showed a decline in Rhot1 and OXPHOS protein levels as well as in insulin secretion and content, silencing of *RHOT1* for 72 h in human islets from non-diabetic donors reduced GSIS and caused mitochondrial dysfunction without changing the insulin content. The discrepancy in insulin content between the two models may be due to that the knock-down specifically focus on the effect of RHOT1, whereas in the GK rat, Rhot1 is probably one of several factors, which together contribute to diabetes. Moreover, we previously found reduced insulin secretion *and* content when comparing islets from human donors with T2D versus non-diabetic controls[9], and in the present study we found reduced *RHOT1* expression in the same setting.

It is debated whether DNA methylation is a cause or consequence of pancreatic islet dysfunction and T2D. Regardless, this study clearly identified T2D candidate genes of importance for β-cell function and insulin secretion, as well as novel epigenetic alterations in islets from patients with T2D. The fact that we found altered DNA methylation that was associated with HbA1c in islets already in non-diabetic individuals and future T2D in blood from a prospective cohort support a causative

contribution to T2D. The epigenetic changes observed in human islets with consequences for insulin secretion may be targeted by future therapies or used as predictive biomarkers. The results will also be used to better understand the epigenetic dysregulation leading to impaired insulin secretion and consequently manifest T2D.

## Methods
### Ethics statement
Written informed consent was obtained from pancreatic donors or their relatives, and all procedures were approved by the Swedish Ethical Review Authority (Permit numbers 2007-05 and 2011-263). The EPIC-Potsdam study procedures were approved by the Ethics Committee of the Medical Association of the State of Brandenburg (Germany) and participants provided written informed consent. The human studies followed the Helsinki Declaration. No participant compensation was given. Animal experiments were performed with permission of the Animal Ethics Committee of Lund University (Permit number 5.8.18-04115/2021) in accordance with the legal requirements of the European Community (86/609/EEC).

### Human cohorts
Human pancreatic islets from multiorgan donors were isolated at The Nordic Network for Islet Transplantation in Uppsala, Sweden. Islets obtained from 100 donors were included in the Islet T2D case-control cohort, 25 of which were diagnosed with T2D and 75 were selected as non-diabetic controls (no T2D diagnosis and HbA1c < 42 mmol/mol, Table 1). Both groups in the Islet T2D case-control cohort were of similar age, and all islet preparations had a purity ≥70%. For the HbA1c analysis, islets from 114 donors with a wide range in HbA1c were included, and none of them were previously diagnosed with diabetes (Islet HbA1c cohort, Table 1). These two "sub-cohorts" include overlapping non-diabetic samples, and islets from a total of 139 pancreatic donors was included in this study. Replication using pyrosequencing was performed in a subset of the Islet T2D case-control cohort, including human islets from 83 donors (Supplementary Data 6).

Blood samples from 540 individuals without T2D at inclusion were obtained from EPIC-Potsdam, a prospective cohort study. We used a nested case-control design with each incident T2D case individually matched to one non-diabetic control according to the following matching criteria: age, sex, fasting time, time of day of blood sampling, and season at blood sampling. Furthermore, each non-diabetic control participant was chosen to have a diabetes-free follow-up time of at least the duration of the follow-up of the respective case. Median time to T2D diagnosis was 3.8 (IQR: 2.0–5.3) years. Recruitment was conducted between 1994 and 1998, and the final

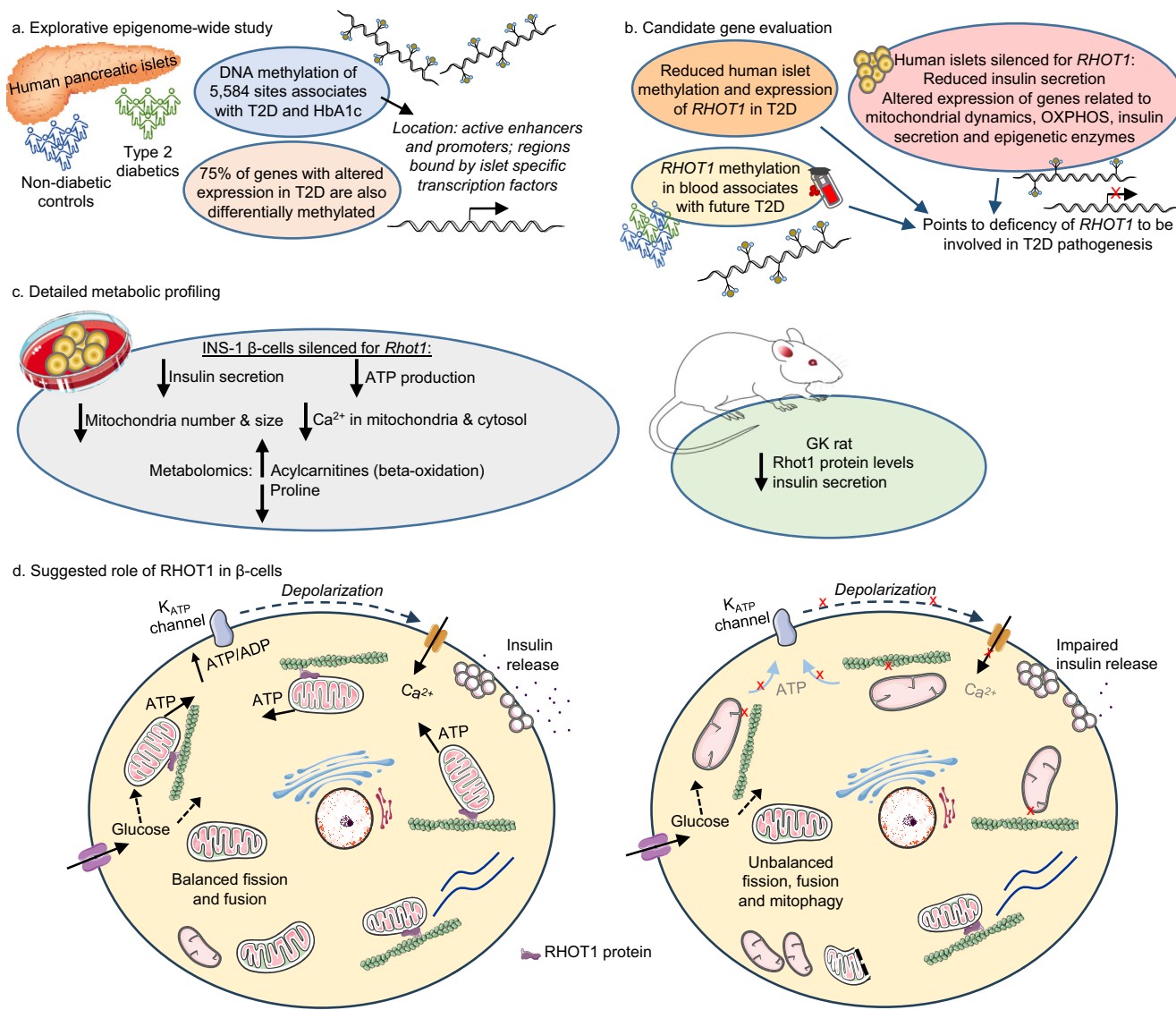

**Fig. 7 | Study summary. a** An explorative epigenome-wide study of ~850,000 DNA methylation sites discovered 5584 sites in human islets whose methylation associated with both T2D in the Islet T2D case-control cohort (25 cases, 75 controls) and HbA1c in the Islet HbA1c cohort (114 donors not previously diagnosed with T2D). Differential DNA methylation was found in regions of importance for gene regulation and in genes differentially expressed in islets from donors with T2D vs. controls. **b** The candidate gene evaluation found that methylation of some sites associates with future T2D development when analyzed in blood ($n = 540$), and that *RHOT1* deficiency in human islets reduces insulin secretion and alters the expression of genes related to mitochondrial function. **c** Metabolic profiling using β-cells silenced for *Rhot1* confirmed reduced insulin secretion and discovered a reduced ATP/ADP ratio, mitochondrial mass, $Ca^{2+}$, and respiration, as well as altered metabolite levels. A rodent T2D model, the GK rat, exhibited RHOT1 deficiency. **d** Based on our results, we suggest that RHOT1 is needed for proper mitochondrial movement in β-cells, creating a balance in mitochondrial dynamics to maintain functional mitochondria. Left: Functional mitochondria with balanced fission, fusion, and mitochondrial movement. Right: Impaired RHOT1 expression may affect mitochondrial movement, with consequences for mitochondrial dynamics (unbalanced fusion/fission processes and altered mitophagy rate). A reduced number of functional mitochondria may impact ATP production and ultimately impair GSIS. This figure was partly generated using Servier Medical Art, provided by Servier, licensed under a Creative Commons Attribution 3.0 unported license.

censoring date was August 2005 (Supplementary Data 13)[58]. Cases were considered when the diagnosis (ICD-10 E11) was confirmed by the last treating physician.

The sex of all donors was determined by self-reporting and concordant genetic testing. Samples from both women and men were analyzed and sex was included as a co-variate in all analyses.

### Primary cell cultures and cell lines
Human pancreatic islets from five non-diabetic donors (obtained according to the section above) were used for functional in vitro studies of candidate genes identified in this study. The donors were three males and two females with an average age of 70.2 years (range 61–81 years), average BMI of 24.3 kg/m² (range 21–30 kg/m²), average

HbA1c of 38.2 mmol/mol (range 34–42 mmol/mol), and islet purity between 95% and 98%. The human pancreatic islets were cultured as described in subsequent sections. For additional in vitro experiments, we used the rat INS-1 832/13 β-cell line[59], kindly provided by Professor Cristopher Newgard, Duke University, Durham, NC, USA. The cells were grown at 37 °C and 5% $CO_2$ in a humidified atmosphere. The culture medium was RPMI-1640 supplemented with 10% fetal calf serum, 10 mM HEPES, 2 mM glutamine, 1 mM sodium pyruvate, and 50 μM beta-mercaptoethanol. INS-1 832/13 β-cell cultures were regularly tested for insulin secretion (see below) and with the Mycoalert Mycoplasma Detection kit (Lonza, Basel, Switzerland) to ensure they were not contaminated with other cell types or mycoplasma, respectively.

## DNA methylation analysis

300 ng DNA extracted from human pancreatic islets ($n = 139$) was bisulfite converted and then analyzed using Infinium MethylationEPIC BeadChips (Illumina, San Diego, CA, USA). The EPIC BeadChip covers ~850,000 CpG sites with comprehensive genome-wide coverage focusing on functional genomic regions. The BeadChips were processed using the Illumina iScan system and raw methylation values for each probe calculated in the GenomeStudio Methylation module (Illumina). QC based on built-in control probes for staining, hybridization, extension and specificity, and the bisulfite conversion efficiency was performed using GenomeStudio.

The DNA methylation data for the human islets were then exported from GenomeStudio and further processed in Bioconductor[60]. Using the lumi package[61], β-values were converted to M-values for statistical analysis[62]. All 139 samples were processed together starting with filtering of probes (removal of 2845 probes with mean detection $p$-value $\geq 0.01$, 148 probes on the Y chromosome, 59 rs probes, 2927 ch probes, and $42,252 + 799$ probes reported to map to several genomic locations or being polymorphic[63]), followed by normalization procedures, background correction, quantile normalization, BMIQ[64], and finally ComBat for removal of batch effects[65]. The final dataset included DNA methylation results for 816,888 CpG sites. For interpretation purposes, data are presented as β-values in all tables and figures. For EPIC-Potsdam, meffil was used to process the idat files. It performs QC, functional normalization, and background correction[66].

## Pathway analysis

For gene set analysis of the genome-wide DNA methylation data, we used the methylGSA Bioconductor package on the full EPIC array data set[15]. We applied the methylRRA function that adjusts for number of CpG sites per gene using robust rank aggregation, followed by gene set enrichment analysis (GSEA). Over-representation of gene sets was related to KEGG pathways (gene set size: 10-500), and FDR was used to correct for multiple testing.

## Pyrosequencing

For technical validation, pyrosequencing assays were designed for four sites annotated to *CDKN1A*, *HDAC4*, *TXNIP*, and *RHOT1* from the EPIC array using PyroMark assay design software 2.0 (Qiagen, Hilden, Germany). The validation was performed in islets (24 T2D cases, 59 controls) based on available islet DNA. Islet DNA ($n = 83$) was bisulfite treated and analyzed on the PyroMark ID96 using the PyroMark PCR kit and PyroMark Gold Q96 reagents according to the recommended protocol (Qiagen). Primer details and sequences for analysis can be found in Supplementary Methods.

## RNA sequencing

RNA was extracted from human pancreatic islets ($n = 97$) using the AllPrep DNA/RNA kit or miRNeasy Mini Kit (Qiagen). Sample preparation of 1 µg high quality RNA was performed using the TruSeq RNA Library Preparation Kit or TruSeq Stranded Total RNA Library Prep, followed by RNA-seq on a HiSeq 2000 or NextSeq 500 (Illumina), respectively. For data preparation and analysis, quality and adapter trimming was achieved using Trim Galore (https://www.bioinformatics.babraham.ac.uk/ projects/trim_galore/), and then Salmon was used for quantification of transcript expression[67]. QC was achieved using fastQC (https://www.bioinformatics.babraham.ac.uk/projects/fastqc/) and multiQC tools[68]. Downstream analysis was achieved using R[69]. R package tximport[70] was used to import transcript-level abundance, estimated counts, and transcript lengths into R using gencode annotation v. 27. This data set was further filtered by transcript type to only include protein-coding genes. Finally, differential expression analysis was performed using DESeq2[71]. RNA-seq data from some of these islet samples have been published[6,72].

## Manipulation of gene expression and analysis of glucose-stimulated insulin secretion in human islets and INS-1 832/13 β-cells

For siRNA transfection in human islets, 100–200 islets were seeded in culture dishes with a final transfection volume of 2.5 mL containing 50 nM of siRNA in Opti-MEM reduced serum media and 6.25 µL of Lipofectamine RNAiMAX (Life Technologies, CA, USA). To improve transfection efficiency, a second transfection was performed 24 h after the first transfection. All functional experiments were performed 72 h after the first transfection. siRNA and TaqMan assays (ThermoFisher Scientific, Waltham, MA, USA) targeting *CABLES1* (siRNA ID s40774, TaqMan assay ID Hs01106667_m1), *FOXP1* (s532467, Hs00908900_m1), *RHOT1* (s30650, Hs00430256_m1), or *TBC1D4* (s19142, Hs00952765_m1) mRNA were used. Gene expression was normalized to the geometric means of two endogenous controls, *PPIA* (Hs04194521_s1) and *HPRT1* (Hs02800695_m1), and relative expression calculated by the ΔΔCt method[73].

Analysis of GSIS in human pancreatic islets started with pre-incubation for 30 min in 2.8 mM glucose, followed by 1 h stimulation with either 2.8 mM glucose (basal secretion) or 16.7 mM glucose (GSIS) in 0.5 mL of KREBS buffer (2.5 mM CaCl₂, 4.7 mM KCl, 120 mM NaCl, 25 mM NaHCO₃, 1.2 mM KH₂PO₄, 1.2 mM MgSO₄, and 10 mM HEPES). Insulin secretion was measured using Mercodia Insulin ELISA #10-1113-01 (human) and the data were normalized to the total insulin content. From each well, protein was extracted using 100 µL RIPA buffer (0.1% SDS, 150 nM NaCl, 1% Triton X-100, 50 mM Tris-Cl, pH 8, and EDTA-free protease inhibitor; Roche, Branchburg, NJ, USA) and analyzed on a TECAN Infinite M200 microplate reader with Magellan software.

For siRNA transfection of INS-1 832/13 β-cells, $2.0 \times 10^5$ cells per well were seeded on 24-well plates and cultured overnight in RPMI-1640 supplemented with 10% fetal calf serum, 10 mM HEPES, 2 mM glutamine, 1 mM sodium pyruvate, and 50 µM beta-mercaptoethanol. Transfection was performed using 25 nM siRNA for *Rhot1* (s235907, Life Technologies, Paisley, UK) and Lipofectamine RNAiMAX (ThermoFisher Scientific). RNA was isolated 72 h after transfection using miRNeasy (Qiagen) and converted to cDNA by the RevertAid First Strand cDNA synthesis kit (ThermoFisher Scientific). Knockdown was verified on the ViiA7 qPCR system (Life Technologies) using the TaqMan assay for *Rhot1* (Rn01751396_m1) and *Hprt1* (Rn01527840_m1) as the endogenous control. Relative expression was calculated by the ΔΔCt method.

For insulin secretion measurements performed 72 h post-transfection, INS-1 832/13 β-cells were washed using HEPES balanced salt solution (HBSS; 114 mM NaCl, 4.7 mM KCl, 1.2 mM KH₂PO₄, 1.16 mM MgSO₄, 20 mM HEPES, 2.5 mM CaCl₂, 25.5 mM NaHCO₃, 0.2% bovine serum albumin, pH 7.2) supplemented with 2.8 mM glucose, followed by 2 h pre-incubation at 37 °C in the same buffer. Insulin secretion was analyzed in static incubations by stimulating the β-cells with fresh HBSS and 2.8 or 16.7 mM glucose for 1 h, with or without the addition of 35 mM K⁺. Insulin released into the buffer was measured by a rat insulin ELISA (Mercodia) and normalized to total insulin content (ELISA) or total amount of protein in each well (BCA Protein Assay Kit; ThermoFisher Scientific).

## Luciferase assay

1540 bp of the *RHOT1* promoter (Supplementary Methods) was inserted upstream of TSS of a CpG-free luciferase reporter vector (pCpGL-basic, kindly provided by Dr Klug and Dr Rehli[74]). SssI, a DNA methyltransferase (New England Biolabs, Ipswich, MA, USA), was then used for methylation of the construct. INS-1 832/13 β-cells were co-transfected with 25 ng methylated pCpGL-vector including the *RHOT1* promoter together with 4 ng of pRL renilla luciferase control reporter vector (Promega, Madison, WI, USA). Firefly luciferase luminescence, as the value of transcriptional activity, was measured using the Dual-Luciferase® Reporter Assay System (Promega) and an

Infinite® M200 PRO multiplate reader (Tecan Group Ltd., Männedorf, Switzerland). Cells transfected with an empty pCpGL-vector were used as background control for firefly luciferase results, and untransfected cells were used as a background for renilla results.

## Gene expression array

Total RNA was extracted from human islets after siRNA experiments (including five donors, all treated with five different siRNAs) using the Qiagen miRNeasy isolation kit according to the manufacturer's recommendations (Qiagen). The 25 RNA samples were analyzed using Clariom S assays (Affymetrix) at BEA, Bioinformatics and Expression analysis core facility, Department of Biosciences and Nutrition, Karolinska Institutet, Huddinge, Sweden. Data were processed using the Transcriptome Analysis Console (TAC) and the SST-RMA analysis method. Limma moderated $t$-tests (paired) were used to detect differences between control islets and islets treated with siRNAs for the four respective genes.

## Western blot analysis

INS-1 832/13 β-cells were lysed in ice-cold RIPA buffer (50 mM Tris-HCl pH 7.4−7.6, 150 mM NaCl, 2 mM ethylenediaminetetraacetic acid [EDTA], 1% Triton X-100, 0.5% Na-deoxycholate, and 0.1% sodium dodecyl sulfate [SDS]) containing protease inhibitor cocktail (Sigma Aldrich, St. Louis, MO, USA) 72 h post-transfection. The lysates were spun at 11,000 g for 3 min to pellet debris and the protein concentration of the supernatant measured by the BCA assay (Thermo-Fisher Scientific). Lysates containing 10 μg of protein were mixed with 5x sample buffer containing 10% β-mercaptoethanol and boiled before being separated by SDS-PAGE on TGX Stain-free gels (Bio-Rad, Hercules, California, USA). The gel was activated in a ChemiDoc MP and the separated proteins transferred to PVDF membranes using the Trans-Blot Turbo System (Bio-Rad). Total protein was imaged in the ChemiDoc MP, and then membranes were blocked in 5% fat free dry milk in TBS-T (Tris buffered saline with 0.05% Tween). Membranes were incubated overnight at 4 °C with primary antibodies diluted in TBS-T containing 5% BSA. The primary antibodies were citrate synthase (#14309 S, clone D7V8B, 1:1000, Cell Signaling Technologies, Danvers, MA, USA), cytochrome C (#4272 T; 1:1000, Cell Signaling Technologies), LAMP1 (ab24170, 1:750, abcam, Cambridge, UK), LC3B (NB100−2220, 1:1000, Novus Biologicals, Centennial, CO, USA), OPA1 (#80471, D6U6N, 1:1000, Cell Signaling Technologies), p62 (ab91526, 1:500, abcam), Parkin (ab77924, clone PRK8, 1:750, abcam), PINK1 (ab23707, 1:750, abcam), RHOT1 (ab188029, clone CL1083, 1:500, abcam), and total OXPHOS rodent antibody cocktail including antibodies against NDUFB8, SDHB, UQCRC2, MTCO1, and ATP5A (#ab110413-MS604, clones 20E9DH10C12, 21A11AE7, 13G12AF12BB11, 1D6E1A8, and 15H4C4, 1:250, abcam). The Rhot1 antibody was validated by siRNA mediated knockdown (Fig. 5c). All antibodies were validated by respective supplier by use of, e.g., knockout cells. After washing in TBS-T, the membranes were incubated with HRP-conjugated secondary antibodies diluted in TBS-T with 5% fat free dry milk for 1 h at room temperature. The secondary antibodies were goat anti-rabbit (#7074, 1:10000, Cell Signaling Technologies) and goat anti-mouse (1706516, 1:5000, Bio-Rad). Signals were developed using the Clarity Western ECL reagent (Bio-Rad) and imaged in a ChemiDoc MP. The antibody-specific signal was quantified and we then used total protein normalization where the detected antibody-specific signal is normalized to total protein in ImageLab software version 6.0.1 (V3 Western Workflow, Bio-Rad). With total protein normalization, the antibody specific signal is normalized to the total amount of protein loaded in each lane. This method is more stable for different loading amounts than using housekeeping genes and avoids the risk of the internal control being affected by experimental conditions[75].

## Mitochondrial staining followed by high-content automated microscopy

INS-1 832/13 β-cells were washed with PBS and stained with MitoTracker Deep Red FM (400 nM; ThermoFisher Scientific, M22426) for 30 min at 37 °C. After mitochondrial staining, β-cells were washed twice with PBS and then fixed with 4% PFA for 15 min. The cells were washed again with PBS and nuclei counterstained with DAPI (1:1000 for 15 min at room temperature). MitoTracker-positive staining was detected using spot detection application on a Cellomics ArrayScan VTI HCS Reader (ThermoFisher Scientific). First, the cytoplasm of the β-cell was defined by a ring surrounding the DAPI-stained nuclei. DAPI-positive nuclei were defined based on intensity, area, and shape. Next, the number and size of MitoTracker spots within the rings were detected. Data from 200 fields per well, with at least 50 valid fields with cells, were obtained. The average number and size of MitoTracker-positive spots were defined per cell for each well. All conditions were performed in triplicate wells for each experiment ($n = 3$), for three different passages, and with 200 fields analyzed from thousands of β-cells each.

## Detection of mitochondrial superoxide production in live β-cells

Transfected INS-1 823/13 β-cells were washed with 300 μL of pre-warmed PBS + 2% fetal calf serum (FCS) and centrifuged for 5 min at 300 g. Cells were resuspended in 50 μL of PBS + 2% FCS and stained with 10 μM MitoSox Red (ThermoFisher Scientific, M36008) for 15 min at 37 °C. After staining the β-cells, 300 μL of PBS + 2% FCS was added to wash them. After centrifuging for 5 min at 300 g, β-cells were resuspended in PBS + 2% FCS and directly acquired in the flow cytometer (CytoFLEX Beckman Coulter).

## ATP measurement in single INS-1 832/13 β-cells

Approximately 70,000 INS-1 832/13 β-cells were seeded on poly-D-lysine (1 mg/mL) coated Lab-Tek chambered cover glass (Thermo-Fisher Scientific). INS-1 832/13 β-cells were transfected with siRhot1 or siNC together with 1 μg of PercevalHR plasmid DNA (Addgene ID: #21737) per well using Lipofectamine 3000 (ThermoFisher Scientific) for 72 h. PercevalHR is a probe that reflects the ATP/ADP ratio of the cell. The β-cells were pre-incubated for 90 min at 37 °C in experimental buffer (pH 7.4): 3.6 mM KCl, 1.3 mM $CaCl_2$, 0.5 mM $MgSO_4$, 0.5 mM $Na_2HPO_4$, 10 mM Hepes, 5 mM $NaHCO_3$, and 135 mM NaCl supplemented with 2.8 mM glucose. The cover glass with adhered cells was mounted on the stage of the microscope (Zeiss Axiovert 200 M, Carl Zeiss AB, Stockholm, Sweden) equipped with a confocal unit. PercevalHR was excited at 488 nm and emission detected at 520 nm.

## Single live INS-1 832/13 β-cell mitochondrial $Ca^{2+}$ measurements

Single β-cell $[Ca^{2+}]_{mito}$ measurements were carried out using genetically encoded mitochondria-targeted $Ca^{2+}$ biosensor mito-case12 (Evrogen cat# FP992; Moscow, Russia). INS-1 832/13 β-cells were seeded onto poly-D-lysine-coated 8-well chambered cover glasses (Lab-Tek) at a cell density of 70,000 cells/cm2. Twenty-four hours after seeding, the β-cells were co-transfected with siRhot1 or siNC and 1 μg of plasmid encoding mito-case12 at around 50% cell confluency for 72 h. On the day of imaging, β-cells were pre-incubated at 37 °C in 400 μL of experimental buffer (3.6 mM KCl, 1.3 mM $CaCl_2$, 0.5 mM $MgSO_4$, 0.5 mM $Na_2HPO_4$, 10 mM Hepes, 5 mM $NaHCO_3$, and 135 mM NaCl; pH 7.4) containing 2.8 mM glucose for 1.5 h. Cells were imaged using a 488 nm laser and emission recorded at 505-530 nm on a Zeiss LSM510 inverted confocal fluorescence microscope using a 100x/1.45 objective. Images were recorded at a frequency of 0.25 Hz (scan time = 3.93 s).

## Oxygen consumption rate measurements

The Seahorse Extracellular Flux Analyzer XF24 (Agilent Technologies, Santa Clara, CA, USA) was used to measure oxygen consumption rate. Approximately 70,000 INS-1 832/13 β-cells were seeded on poly-D-

lysine (1 mg/mL) coated XF24 24-well cell culture microplates in 100 μL complete RPMI 1640 medium without antibiotics. The β-cells were transfected with siRhot1 or siNC as described above and the assay performed 72 h post-transfection. Using the Wave software (v2.6.1.53; Agilent Technologies), the experiments were designed to determine respiration in low (2.8 mM) glucose conditions, and then transition to high (16.7 mM) glucose. The proportion of respiration driving ATP synthesis and proton leak were determined by the addition of oligomycin (4 μg/mL). Afterwards, 4 μM of the uncoupler carbonyl cyanide-p-trifluoromethoxyphenylhydrazone (FCCP) was added to determine the maximal respiratory capacity. Finally, 1 μM rotenone and antimycin A was added to block the transfer of electrons from complex I to ubiquinone and through complex III, respectively.

## Cytoplasmic Ca$^{2+}$ measurements

INS-1 832/13 β-cells were seeded onto poly-D-lysine (1 μg/mL) coated 8-well chambered cover glasses (Lab-Tek, Naperville, IL, USA) at a density of 70,000 cells per well. Twenty-four hours after seeding, INS-1 832/13 β-cells were transfected with siRNA (siRhot1 or siNC) as described above and incubated for another 72 h prior to imaging. On the day of imaging, β-cells were pre-incubated at 37 °C in 400 μL of imaging buffer (135 mM NaCl, 3.6 mM KCl, 1.5 mM CaCl$_2$, 0.5 mM MgSO$_4$, 0.5 mM Na$_2$HPO$_4$, 10 mM HEPES, 5 mM NaHCO$_3$, pH 7.4) containing 2.8 mM glucose. After incubation for 1.5 h, cells were loaded with 2 μM of Fluo4 AM and the incubation continued for an additional 30 min. Immediately prior to imaging, the Fluo4-containing imaging buffer was exchanged with 400 μL of fresh imaging buffer. Imaging was performed using a Zeiss LSM510 inverted confocal fluorescence microscope with a 40x/0.75 objective. Fluo4 AM was excited at 488 nm with emission collected using a 505 nm emission filter. Images were recorded at a frequency of 0.64 Hz (scan time = 1.57 s) using a pinhole diameter of 463 μm. Regions of interest were selected manually using open-source Fiji software. Data were quantified by measuring the area under the curve (AUC) of peaks during the high glucose condition (16.7 mM), whereby peaks were detected at a threshold of a 5% increase from the baseline and required at least two adjacent points.

## Metabolomics

Metabolomic analysis was performed in INS-1 832/13 β-cells transfected with siRhot1 or siNC as described above. β-cells were then collected 72 h post-transfection, including seven different cell passages for each siRNA (*n* = 7). All experiments were performed at both low (2.8 mM) and high (16.7 mM) glucose levels.

**Sample preparation.** Frozen samples were stored at −80 °C for further use. Metabolic profiling by gas chromatography-mass spectrometry (GC-MS) and liquid chromatography-mass spectrometry (LC-MS) was performed at the Swedish Metabolomics Centre (SMC) in Umeå, Sweden. Prior to analysis, the metabolites were extracted as follows. To each sample, 100 μL 90% MeOH and a tungsten bead were added with the following internal standards: 13C3-caffeine, D-sucrose−13C12, 13C9-phenylalanine, D4-cholic acid, salicylic acid-D6, 13C9-caffeic acid, succinic acid-D4, L-glutamic acid-13C5,15 N, putrescine-D4, hexadecanoic acid-13C4, D-glucose-13C6 (obtained from Merck Sigma-Aldrich, St. Louis, MO, USA), L-proline-13C5, alpha-ketoglutarate-13C4, myristic acid-13C3, and cholesterol-D7 (from Cil, Andover, MA, USA). The sample was shaken at 30 Hz for 3 min in a mixer mill and then centrifuged at 18,620 g at 4 °C for 10 min. The supernatant, 200 μL for LC-MS analysis and 50 μL for GC-MS analysis, was transferred to micro vials and evaporated to dryness in a speed-vac concentrator. Solvents were evaporated and the samples stored at −80 °C until analysis. Aliquots of the remaining supernatants were pooled and used to create QC samples. MS-MS analysis (LC-MS) was run on the QC samples for identification purposes. The samples were analyzed in batches according to a randomized run order on both GC-MS and LC-MS.

**GC-MS analysis.** The samples were derivatized prior to analysis. In detail, 10 μL of methoxyamine (15 μg/μL in pyridine) was added to the dry sample and the sample shaken vigorously for 10 min before left to react at room temperature for 16 h. Next, 20 μL of a 1:1 mixture of MSTFA (1%TMCS) and heptane (containing 7.5 ng/μL methyl stearate) were added to the shaken sample and left to react for 1 h at room temperature. A 1 μL aliquot of the derivatized sample was injected in splitless mode by an L-PAL3 autosampler (CTC Analytics AG, Switzerland) into an Agilent 7890B gas chromatograph equipped with a 10 m x 0.18 mm fused silica capillary column with a chemically bonded 0.18 μm Rxi-5 Sil MS stationary phase (Restek Corporation, U.S.) The injector temperature was 270 °C, the purge flow rate 20 mL/min, and the purge was turned on after 60 s. The gas flow rate through the column was 1 mL/min. The column temperature was held at 70 °C for 2 min, and then increased by 40 °C/min to 320 °C and held there for 2 min. The column effluent was introduced into the ion source of a Pegasus BT time-of-flight mass spectrometer, GC/TOFMS (Leco Corp., St Joseph, MI, USA). The transfer line and ion source temperatures were 250 °C and 200 °C, respectively. Ions were generated by a 70 eV electron beam at an ionization current of 2.0 mA, and 30 spectra were recorded per second in the mass range m/z 50 to 800. The acceleration voltage was turned on after a solvent delay of 150 s. The detector voltage was 1800–2300 V.

**LC-MS analysis.** Before LC-MS analysis, the sample was resuspended in 10 + 10 μL methanol and water. Each batch of samples was analyzed first in positive mode. After all samples within a batch had been analyzed, the instrument was switched to negative mode and a second injection of each sample achieved. The chromatographic separation was performed on an Agilent 1290 Infinity UHPLC system (Agilent Technologies, Waldbronn, Germany). A total of 2 μL of each sample was injected onto an Acquity UPLC HSS T3, 2.1 x 50 mm, 1.8 μm C18 column in combination with a 2.1 mm x 5 mm, 1.8 μm VanGuard pre-column (Waters Corporation, Milford, MA, USA) held at 40 °C. The gradient elution buffers were H$_2$O and 0.1% formic acid (A) and 75/25 acetonitrile:2-propanol and 0.1% formic acid (B). The flow rate was 0.5 mL/min. The compounds were eluted with a linear gradient consisting of 0.1–10% B over 2 min. B was increased to 99% over 5 min and held at 99% for 2 min before being decreased to 0.1% for 0.3 min and the flow rate increased to 0.8 mL/min for 0.5 min. These conditions were held for 0.9 min, after which the flow rate was reduced to 0.5 mL/min for 0.1 min before the next injection. The compounds were detected by an Agilent 6546 Q-TOF mass spectrometer equipped with a jet stream electrospray ion source operating in positive or negative ion mode. The settings were kept identical between the modes with the exception of the capillary voltage. A reference interface was connected for accurate mass measurements; the reference ions purine (4 μM) and HP-0921 (Hexakis(1H, 1H, 3H-tetrafluoropropoxy)phosphazine) (1 μM, Agilent Technologies) were infused directly into the MS at a flow rate of 0.05 mL/min for internal calibration. The monitored ions were purine m/z 121.05 and m/z 119.03632, and HP-0921 m/z 922.0098 and m/z 966.000725 for positive and negative mode, respectively. The gas temperature was set to 150 °C, the drying gas flow to 8 L/min, and the nebulizer pressure to 35 psig. The sheath gas temperature was set to 350 °C and the sheath gas flow 11 L/min. The capillary voltage was set to 4000 V in positive ion mode, and to 4000 V in negative ion mode. The nozzle voltage was 300 V. The fragmentor voltage was 120 V, the skimmer 65 V, and the OCT 1 RF Vpp 750 V. The collision energy was set to 0 V. The m/z range was 70–1700, and data were collected in centroid mode with an acquisition rate of 4 scans/s (1977 transients/spectrum).

**Solvents.** Methanol, HPLC grade was obtained from Fischer Scientific (Waltham, MA, USA). Chloroform, Suprasolv for GC was obtained from Merck (Darmstadt, Germany). Acetonitrile, HPLC grade was obtained

from Fischer Scientific (Waltham, MA, USA). 2-Propanol, HPLC grade was obtained from VWR (Radnor, PA, USA). $H_2O$ was purified by Milli-Q.

## Diabetic animal model

Animal experiments were performed with permission of the Animal Ethics Committee of Lund University (Permit number 5.8.18-04115/2021) in accordance with the legal requirements of the European Community (86/609/EEC). Male GK rats, developed by selective breeding of Wistar rats[44], and control Wistar rats (Janvier labs, France) were kept in standard 12 h light-dark cycle and given standard chow and water ad libitum. The animals were used at 8 or 12 weeks of age (young and adult GK, respectively). Only male rats were included to reduce the number of animals used.

For Western blot analysis, rat islets were lysed in RIPA buffer and the analysis performed as described above for INS-1 832/13 β-cells using the RHOT1 primary antibody (ab188029, clone CL108, 1:500, abcam), Citrate Synthase antibody (#14309 S, clone D7V8B, 1:1000, Cell Signaling Technologies) and total OXPHOS rodent antibody cocktail including antibodies against NDUFB8, SDHB, UQCRC2, MTCO1, and ATP5A (ab110413-MS604, clones 20E9DH10C12, 21A11AE7, 13G12AF12BB11, 1D6E1A8, and 15H4C4 1:250, abcam).

For the intravenous glucose tolerance test, an intraperitoneal injection of glucose (2 g/kg) was given after a 4 h fast. Glucose levels were measured from the tail vein using a glucometer (ACCU-CHEK Aviva, Roche) at 5, 10, 15, 30, 60, and 120 min.

Pancreatic islets were isolated from GK and control Wistar rats by collagenase digestion and the islets handpicked in cold Hank's buffer with 1 mg/mL bovine serum albumin (BSA) prior to insulin secretion assays. For the insulin secretion assay, batches of 6–10 islets from either GK or Wistar rats were seeded in 24-well plates in quadruplicate and pre-incubated for 30 min in 2.8 mM glucose. Islets were then stimulated for 1 h in either 2.8 mM glucose (basal insulin secretion) or 16.7 mM glucose (GSIS) in 0.5 mL of KREBS buffer. Insulin released into the buffer was measured by a rat insulin ELISA (Mercodia) and normalized to the number of islets.

## Statistical methods

Clinical characteristics between individuals with T2D and non-diabetic controls were analyzed using two-sample, two-tailed t-tests. Differences in the DNA methylation of islets from individuals with T2D compared with controls and the association between islet DNA methylation and HbA1c were analyzed using linear regression adjusted for age, BMI, sex, islet purity, and days in culture. FDR analysis[76] was used to account for multiple testing, and FDR below 5% ($q < 0.05$) was applied. For pyrosequencing, islet methylation data were analyzed using SPSS based on the same linear regression model as described above. We also analyzed the DNA methylation array data using the same linear model, including adjustments for cell composition[11]. To integrate epigenetic information across different methylation sites, we calculated a weighted combined MRS for each gene with altered expression and more than 5 differentially methylated CpG sites within or near (±10 kb) that gene (27 in total). These weighted MRSs were calculated as the sum of the methylation values (%) at each included CpG site in the gene, weighted by CpG-specific effect size (multiplying by β-coefficients from the multiple linear regressions, explained above, when assessing differences in methylation between T2D cases compared with controls)[50]. Then, multiple linear and logistic regression models adjusted for age, sex, BMI, islet purity and days in culture were performed to assess the association between MRSs and T2D.

A nested case-control design was used for the EPIC-Potsdam prospective cohort; thus, calculated estimates are odds ratios from a conditional logistic regression. The model was adjusted for age, sex, waist circumference, smoking, alcohol intake, physical activity, estimated cell proportions[77], and batch effects (smartSVA)[78].

Paired t-tests (two-tailed) were used in silencing experiments comparing siRNAs of target genes and siNC cells.

The non-parametric Mann-Whitney U test was performed in Graphpad Prism for the MitoTracker Staining experiment in siRhot1 vs. siNC β-cells.

For cytoplasmic $Ca^{2+}$ measurements, the statistical analysis was performed using Graphpad Prism, with significance between groups calculated using the non-parametric Mann-Whitney U test.

For the GC-MS data, all non-processed MS-files from the metabolic analysis were exported from ChromaTOF software in NetCDF format, and all pre-treatment procedures, such as baseline correction, chromatogram alignment, data compression, peak-picking, and multivariate curve resolution, were performed in MATLAB (Mathworks, Natick, MA, USA). The extracted mass spectra were identified by comparing their retention index and mass spectra with libraries of retention time indices and mass spectra. The mass spectra and retention index comparisons were performed using NIST MS 2.2 software. Annotation of mass spectra was based on reverse and forward searches in the library.

For the LC-MS data, all data processing was performed using Agilent Masshunter Profinder version B.10.00 (Agilent Technologies). The processing was performed in a targeted fashion using the Batch Targeted feature extraction in Masshunter Profinder. An in-house LC-MS library built up by authentic standards run on the same system with the same chromatographic and mass-spec settings was used as the library for targeted processing.

Two-sided t-tests were used to compare data in adult GK vs. Wistar controls, as well as in adult vs. young GK rats.

## Figure preparation

Graphs in Figs. 1–6 and Supplementary Figs. 1–7 were created using GraphPad Prism. Euler diagrams for Figs. 1g and 4a were created using eulerr (Larsson J, 2019. eulerr: Area-Proportional Euler and Venn Diagrams with Ellipses. R package version 6.0.0, https://cran.r-project.org/package=eulerr) and the Venn diagram for Figs. 4b and 6a using an online tool (http://bioinformatics.psb.ugent.be/webtools/Venn/). Parts of Figs. 1a and 7 were drawn by using pictures from Servier Medical Art, by Servier, licensed under a Creative Commons Attribution 3.0 Unported License (https://creativecommons.org/licenses/by/3.0/).

## Reporting summary

Further information on research design is available in the Nature Portfolio Reporting Summary linked to this article.

# Data availability

The human islet DNA methylation and RNA-seq datasets generated for this study (EPIC DNA methylation data, accession numbers LUDC2022.05.011, LUDC2022.05.012 and RNA-seq, accession number LUDC2022.05.013) were deposited in the LUDC repository (https://www.ludc.lu.se/resources/repository). Data are available upon request through the repository portal. Individual-level data from the human pancreatic islets are not publicly available due to ethical and legal restrictions related to the Swedish Biobanks in Medical Care Act, the Personal Data Act and European Union's General Data Protection Regulation and Data Protection Act. Source data are provided with this paper.

# Code availability

The computer code used to generate the results described in the methods section under DNA methylation analysis and Statistical methods and more details are available from the corresponding author on request.

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

## Acknowledgements

We thank the Nordic Network for Clinical Islet Transplantation (JDRF award 31-2008-413), Human Tissue Laboratory in EXODIAB/Lund University Diabetes Centre, Bioinformatics and Expression Analysis (BEA, supported by the Board of Research at the Karolinska Institute and the Research Committee at Karolinska Hospital) for microarray analysis, the Swedish Metabolomics Centre, Umeå University for metabolic profiling, Claes Wollheim and David Nicholls for valuable discussions, and Anna Hammarberg at MultiPark Cellomics Platform and Eugenia Cordero for technical assistance. EPIC array analysis was performed by the SNP&SEQ Technology Platform in Uppsala. The facility is part of the National Genomics Infrastructure (NGI) Sweden and Science for Life Laboratory. The SNP&SEQ Platform is also supported by the Swedish Research Council and the Knut and Alice Wallenberg Foundation. This study was supported by the Swedish Research Council (Dnr 2016-02486, 2018-02567, 2019-01406, 2014-2775, 2018-02435, and 2021-00628), Region Skåne, Strategic Research Area Exodiab (Dnr 2009-1039), the Novo Nordisk Foundation, the Swedish Foundation for Strategic Research (Dnr IRC15-0067), The European Research Council (ERC-Paintbox), the Diabetes Foundation, Kungliga Fysiografiska Sällskapet i Lund, the German Federal Ministry of Education and Research, and the

State of Brandenburg to the German Center for Diabetes Research (82DZD00302, 82DZD03D03), as well as the Syskonen Svensson, Magnus Bergvall, Åke Wiberg, and Påhlsson Foundations.

## Author contributions

C.L. initiated the project. All authors designed experiments and/or analyses. T.R., J.K.O., A.P., A.H., K.P., F.E., S.GC., A.K., H.S., A.W., P.V., M.B.S., S.R., and K.B. performed experiments and/or analyses. H.M. and L.E. designed experiments and interpreted data. All authors analyzed and interpreted the data. T.R. and C.L. drafted the manuscript. All authors read and edited the manuscript.

## Funding

## Competing interests

The authors declare no competing interests.
