## [Peer Review File · Nature Communications]

Genes with epigenetic alterations in human pancreatic islets impact mitochondrial function, insulin secretion, and type 2 diabetesEditorial Note: Parts of this Peer Review File have been redacted as indicated to remove third-party material where no permission to publish could be obtained.

REVIEWER COMMENTS

Reviewer #1 (Remarks to the Author):

In the manuscript "Genes with epigenetic alterations in human pancreatic islets impact mitochondrial function, insulin secretion, and type 2 diabetes" by T. Rönn and co-workers, the Authors aim better understand T2D pathogenesis and dissect the mechanisms that perturb islet insulin secretion. To this end, they have adopted an epigenome-wide association study in pancreatic islets from a larger T2D case-control cohort, testing if there was an association with HbA1c, using islets of individuals not previously diagnosed with diabetes. Next, they integrated the results with islet data of expression, open chromatin regions, transcription factor binding occupancy, and histone modifications. Moreover, they performed in vitro approaches to examine the effects of candidate genes in human islets and clonal β -cells to dissect their effects on cell metabolism and insulin secretion, validating also the findings in a rodent T2D model. It is shown that novel DNA methylation alterations impact T2D progression. Working both on different levels of epigenetic marks and transcriptional activity and on cellular function and metabolism in human islets, clonal β -cells, and a diabetic rat model, RHOT1 was identified, as a key regulator in islet function and DNA methylation of RHOT1 in blood also predicted T2D.

This is an exciting and very timely manuscript reporting novel data on epigenetic dysregulation in T2D that is well-established but unfortunately still unexplained.

The manuscript is well-written and the arguments are easy to follow. All data presented are convincing, not overinterpreted, and support the conclusion.

A major strength is the different and complementary approaches that make the manuscript solid and certainly quite unique. The methods are well-detailed data analysis is well performed as represented by adjustment for cell composition.

The following comments are provided to the Authors in a fully collaborative spirit, as an attempt to contribute to further improving this nice study.

Only some minor concerns are required:

1. In the Introduction, to highlight the novelty of the work and knowledge added to bridge the gap in this field of study, it should better:

- argue the knowledge gained to date on epigenetic dysregulation and T2D and,
- discussing the importance to establish if the epigenetic alterations predispose to disease.

2. The star of significance should be larger in:

- In Figures 2a (only in FOXP1 RNA-Seq panel), b, c, and d;
- in Figures 3g and h

3. figure 6 should be improved because of not very interpretive.

Reviewer #2 (Remarks to the Author):

The manuscript by Rönn et al. found that alteration of methylation in human islets associated with HbA1c in individuals not diagnosed with T2D. Importantly, RHOT1, as a key regulator of insulin secretion in human islets, was related to insulin secretion, ATP/ADP ratio, mitochondrial mass, Ca²⁺ and respiration. Meanwhile, RHOT1-deficiency also regulated mitochondrial dynamics and metabolites, including L-proline, glycine, GABA and carnitines. Notably, RHOT1 methylation in blood associates with future T2D. Overall, although the results of this study are interesting, but several important concerns need to be carefully addressed.

1. In the manuscript, the authors found that lower RHOT1 levels in β -cells led to deregulation of mitophagy and altered expression of proteins involved in mitochondrial dynamics and metabolism, resulting in mitochondrial dysfunction and perturbed GSIS. What are the possible mechanisms by which RHOT1 maintains mitochondrial homeostasis? The authors should discuss this issue in the discussion section.

2. RHOT1-deficiency in β -cells causes reduced insulin secretion. How does RHOT1 regulate insulin secretion in β -cells? Please explain that.

3. The authors found an almost complete ablation of Rhot1 protein levels in islets of adult GK rats

with the decrease of insulin secretion. Therefore, what is the relationship between Rhot1 expression and insulin secretion in GK rats. It would be helpful to analyze the correlation between Rhot1 expression and insulin secretion.

4. The authors found the important role of RHOT1 in maintaining mitochondrial function, but there was no data related to the mitochondrial function in GK rats. It would be better to provide the related results to analyze the relationship between Rhot1 expression and mitochondrial function in GK rats.

5. In Figure 3, RHOT1 expression was decreased by all four siRNA in human islets, such as FOXP1, TBC1D4, RHOT1 and CABLES1. Therefore, how does FOXP1, TBC1D4 and CABLES1 regulate the expression of RHOT1.

6. In figure 4c and 5g, there was no GAPDH or ACTIN. How does the authors perform the quantification in WB analysis. Please explain that in detail.

7. In some Figure legends, the sample size n was missed.

Reviewer #3 (Remarks to the Author):

The manuscript presents a genome-wide DNA methylation study comparing healthy (75 samples) and T2D (25 samples) human islets. The authors also examine the correlation between differential DNA methylation and HbA1C using islets from an independent cohort (114 samples). They integrate the methylation data with RNA-seq, ATAC-seq, transcription factor and histone modifications' ChIP-seq data. They reported four T2D-associated methylation sites associated with lower risk of T2D in blood samples. Selected candidate genes are validated using siRNA knockdown in a rat beta cell line. The study highlights the potential role of Rhot1 in regulating mitochondria function and metabolites, as well as its deficiency in a rodent diabetes model associated with reduced insulin content.

Although this study has multiple points, the logical flow between them is not coherent or compelling. Although the authors hint a mechanistic connection between DNA methylation and gene expression, the evidence is preliminary. A lot of results are only described in the text with a brief reference to an excel file without proper presentation or quantitative analysis. A key result about the prospective T2D risk analysis not adequately presented or discussed. Overall, I feel that this manuscript has major flaws and at least needs a lot of improvements to be acceptable for publication in NC. Here are some specific points.

1. I highly recommend including a panel in figure 1 (such as a heatmap) showing the changes the DNA methylation changes in every individual. This will help readers understand what kind of changes we are looking at, and how consistent the changes are between the control and T2D groups. Without a global picture, the readers may feel that the authors are only cherry-picking some best examples.

2. The p-values for the enrichment in Figure 1e is missing. The significance of the enrichment is not clear.

3. The authors claimed some consistency between current study and previous studies (Dayeh et al.; Volkov et al.) but provide very little details. Only an excel file is provided. Summarization of the comparison should be included. For example, how many hyper-methylated sites from this study are hyper or hypo- methylated in the other studies; conversely, how many hypo-methylated sites from this study are hyper or hypo- methylated in the other studies. Quantitative statistical analysis should be also included.

4. The changes of DNA methylation shown in Fig. 2 examples all very subtle, making me wonder if these changes are meaningful. Figures 2a-2d and S2a-2c would benefit from being shown in a genome browser with larger range, with the information of other CpG sites and the results of nearby genes.

5. It is unclear whether the differential methylation sites are only enriched in the DEG genes and how significant this is.

6. The section "DNA methylation associates with future T2D" is extremely brief. Again, the authors only refer the readers to an excel file. Here the authors test the putative CpG sites in blood samples. Should we worry about the consistency between islets and blood? This should be at least discussed. In fact, the authors could have screened for candidate predictive DNA methylation sites from blood samples in the beginning. Wouldn't that give better outcomes? The authors names 4 predictive sites, including one with RHOT1, with low T2D risk and low DNA methylation. But in Fig. 1, the RHOT1 CpG show low DNA methylation among T2D patients (i.e., low DNA methylation is associated with increased T2D risk). This is contradictory.

7. Fig. 5g-k shows that in diabetic rat, the down regulation of Rhot1 is correlated with lower insulin content in the islets, suggesting a causal role. However, in human islets (Fig. S4a), siRHOT1 does not affect islet insulin content. This raise a serious question if and how much of the Rhot1 functional results from rat are conserved in human islets.

Author point-by-point response to reviewers' comments

Manuscript: “*Genes with epigenetic alterations in human pancreatic islets impact mitochondrial function, insulin secretion, and type 2 diabetes*” (NCOMMS-23-08677-T)

REVIEWER COMMENTS

Reviewer #1 (Remarks to the Author):

In the manuscript “Genes with epigenetic alterations in human pancreatic islets impact mitochondrial function, insulin secretion, and type 2 diabetes” by T. Rönn and co-workers, the Authors aim better understand T2D pathogenesis and dissect the mechanisms that perturb islet insulin secretion. To this end, they have adopted an epigenome-wide association study in pancreatic islets from a larger T2D case-control cohort, testing if there was an association with HbA1c, using islets of individuals not previously diagnosed with diabetes. Next, they integrated the results with islet data of expression, open chromatin regions, transcription factor binding occupancy, and histone modifications. Moreover, they performed in vitro approaches to examine the effects of candidate genes in human islets and clonal β -cells to dissect their effects on cell metabolism and insulin secretion, validating also the findings in a rodent T2D model. It is shown that novel DNA methylation alterations impact T2D progression. Working both on different levels of epigenetic marks and transcriptional activity and on cellular function and metabolism in human islets, clonal β -cells, and a diabetic rat model, RHOT1 was identified, as a key regulator in islet function and DNA methylation of RHOT1 in blood also predicted T2D.

This is an exciting and very timely manuscript reporting novel data on epigenetic dysregulation in T2D that is well-established but unfortunately still unexplained. The manuscript is well-written and the arguments are easy to follow. All data presented are convincing, not overinterpreted, and support the conclusion.

A major strength is the different and complementary approaches that make the manuscript solid and certainly quite unique. The methods are well-detailed data analysis is well performed as represented by adjustment for cell composition.

The following comments are provided to the Authors in a fully collaborative spirit, as an attempt to contribute to further improving this nice study.

Response: We want to thank Reviewer #1 for reviewing our manuscript and we are happy that our paper was well received and described as well-written, timely and unique. We appreciate the review comments, which further helped us improve our study and we have addressed each concern according to the points below:

Point-by-point response to specific comments:

Only some minor concerns are required:

1. In the Introduction, to highlight the novelty of the work and knowledge added to bridge the gap in this field of study, it should better:

- argue the knowledge gained to date on epigenetic dysregulation and T2D and,
- discussing the importance to establish if the epigenetic alterations predispose to disease.

Response: Based on this valid comment, we added the following information in the introduction on page 2, line 43-60: of the revised ms:

“Epigenetic dysregulation can lead to disease, and there is evidence of associations between the epigenome and T2D²⁻³. Smaller cohorts have shown that DNA methylation is different in human pancreatic islets from T2D cases versus non-diabetic controls⁴⁻⁶. Functional experiments in cultured β -cells of identified candidate genes showing both differential DNA methylation and expression in islets from donors with T2D (e.g., CDKN1A, PDE7B, PARK2 and SOCS2) further linked epigenetic dysregulation in pancreatic islets to impaired insulin secretion^{4,6}. Moreover, increased islet DNA methylation of T2D candidate genes, such as INS, PDX1, GLP1R and PPARGC1A, has also been associated with reduced expression of said genes and abrogated insulin secretion, further supporting a key role of epigenetic dysregulation in pancreatic islets from individuals with diabetes⁷⁻¹⁰. However, though these case-control studies identified differential DNA methylation of numerous genes in pancreatic islets from donors with T2D, they did not establish whether the identified epigenetic alterations predispose an individual to disease. Therefore, studies testing whether epigenetics predispose to T2D and whether epigenetic alterations can be found already in islets from individuals at risk for T2D are desirable. Such studies could provide support that epigenetic alterations in pancreatic islets may cause diabetes rather than being a consequence of the disease. In addition, if epigenetic alterations predispose an individual to diabetes, this information can be used for preventive care, delaying disease onset and progression, and reducing long-term complications and patient suffering. Moreover, larger case-control studies that robustly link epigenetics to T2D are warranted.”

2. The star of significance should be larger in:

- In Figures 2a (only in FOXP1 RNA-Seq panel), b, c, and d;
- in Figures 3g and h

Response: Thanks for this detailed scrutiny. Stars of significance have been changed according to the recommendations (now re-numbered as **Fig. 3a-d, 4g-h**). Please find new versions of all the figures being submitted.

3. figure 6 should be improved because of not very interpretive.

Response: Based on this valid comment, Fig. 6 (now **Fig. 7**) has been improved. We included this figure to summarize the main results and take-home message of this extensive study. As such, it is important that the figure is easy to understand. To improve this figure in accordance with this review comment, we have now changed the layout and structure, and also added a section where we summarize in a cartoon how we, based on our studies in β -cells and previous knowledge based on other cell types, view the normal function of RHOT1 in β -cells and how RHOT1 deficiency may lead to mitochondrial dysfunction and subsequently reduced insulin secretion. Please see **Fig. 7** (previous Fig. 6) in the re-submitted version. Also, the Figure legend has been re-structured and made more informative to make the figure easier to interpret: page 30, line 1169-1185:

“Figure 7. Study summary. a) An explorative epigenome-wide study of ~850,000 DNA methylation sites discovered 5,584 sites in human islets whose methylation associated with both T2D in the Islet T2D case-control cohort (25 cases, 75 controls) and HbA1c in the Islet HbA1c cohort (114 donors not previously diagnosed with T2D). Differential DNA methylation was found in regions of importance for gene regulation, and in genes differentially expressed in islets from donors with T2D vs. non-diabetic controls. b) The candidate gene evaluation found that methylation of some sites predicts T2D development when analyzed in blood (n=450), and that RHOT1 deficiency in human islets reduces insulin secretion and alters the expression of genes related to mitochondrial function. c) Metabolic profiling using β -cells silenced for Rhot1 confirmed reduced insulin secretion and discovered a reduced ATP/ADP ratio, mitochondrial mass, Ca²⁺ and respiration, as well as altered metabolite levels. A rodent T2D model, the GK rat, exhibited RHOT1 deficiency. d) Based on our results, we suggest that RHOT1 is needed for proper mitochondrial movement in β -cells, creating a balance in mitochondrial dynamics

to maintain functional mitochondria. Left: Functional mitochondria with balanced fission, fusion, and mitochondrial movement. Right: Impaired RHOT1 expression may affect mitochondrial movement with consequences for mitochondrial dynamics (unbalanced fusion/fission processes and altered mitophagy rate). A reduced number of functional mitochondria may impact ATP production and ultimately impair GSIS.”

Reviewer #2 (Remarks to the Author):

The manuscript by Rönn et al. found that alteration of methylation in human islets associated with HbA1c in individuals not diagnosed with T2D. Importantly, RHOT1, as a key regulator of insulin secretion in human islets, was related to insulin secretion, ATP/ADP ratio, mitochondrial mass, Ca²⁺ and respiration. Meanwhile, RHOT1-deficiency also regulated mitochondrial dynamics and metabolites, including L-proline, glycine, GABA and carnitines. Notably, RHOT1 methylation in blood associates with future T2D. Overall, although the results of this study are interesting, but several important concerns need to be carefully addressed.

Response: We are grateful for the thorough revision by Reviewer #2 that has helped us improve our manuscript and we are very pleased that also this reviewer believes our study is interesting.

Point-by-point response to specific comments:

1. In the manuscript, the authors found that lower RHOT1 levels in β -cells led to deregulation of mitophagy and altered expression of proteins involved in mitochondrial dynamics and metabolism, resulting in mitochondrial dysfunction and perturbed GSIS. What are the possible mechanisms by which RHOT1 maintains mitochondrial homeostasis? The authors should discuss this issue in the discussion section.

Response: We appreciate this comment. The current knowledge of RHOT1 in pancreatic islets and β -cells is limited. Hence, we performed this extensive study of mitochondrial function and metabolomics in β -cells silenced for *Rhot1*. Based on studies in neurons, Rhot1 (also called Miro1) is attached to the mitochondrial surface, where it connects to the motor proteins kinesin and dynein, which in turn connects to the axon filaments. Thus, Rhot1 is part of a motor/adaptor complex connecting the mitochondria to the axon filaments, and thereby important for mitochondrial movement (ref Mitochondrial Trafficking in Neurons, 2013, Schwarz). Our results from β -cells in combination with this knowledge, suggest that RHOT1 is needed for proper mitochondrial movement, thus creating a balance between fusion, fission and mitophagy, to maintain functional mitochondria. With impaired/reduced RHOT1 expression, mitochondrial movement is affected, with consequences for mitochondrial dynamics (unbalanced fusion/fission processes and altered mitophagy rate), which impact the number of functional mitochondria, resulting in reduced ATP production and ultimately lower glucose-stimulated insulin secretion.

This has now been included in the Discussion, page 12-13, line 467-473: *“Studies in neurons have shown that RHOT1 connects the mitochondria via motor proteins to the axon filaments, thereby regulating mitochondrial movement (Schwarz 2013). Therefore, it is likely that RHOT1 is needed for proper mitochondrial movement also in human islets and β -cells, creating a balance in mitochondrial dynamics and maintaining functional mitochondria, as required to maintain the proper ATP production needed for GSIS. Thus, impaired RHOT1 expression alters mitochondrial dynamics, which impacts the number of functional mitochondria, resulting in reduced ATP production and insulin secretion (Fig. 7d).”*

We have also included a new Fig. 7 with a schematic drawing of RHOT1 function in normal and diseased β -cells (**Fig. 7d**). Page 30, line 1179-1185, Fig. 7d legend: *“Based on our results, we suggest that RHOT1 is needed for proper mitochondrial movement in β -cells, creating a balance in mitochondrial*

dynamics to maintain functional mitochondria. Left: Functional mitochondria with balanced fission, fusion, and mitochondrial movement. Right: Impaired RHOT1 expression may affect mitochondrial movement with consequences for mitochondrial dynamics (unbalanced fusion/fission processes and altered mitophagy rate). A reduced number of functional mitochondria may impact ATP production and ultimately impair GSIS.”

2. RHOT1-deficiency in β -cells causes reduced insulin secretion. How does RHOT1 regulate insulin secretion in β -cells? Please explain that.

Response: RHOT1 is needed for mitochondrial trafficking, hence deficiency leads to altered mitochondrial dynamics and eventually reduced number of functional mitochondria. Functional mitochondria are required to maintain proper ATP production, which in turn, is needed for glucose-stimulated insulin secretion to occur. Please also see the response to comment 1, the inclusion of this to the Discussion part and the revised Fig. 7.

Discussion, page 12-13, line 467-473: *“Studies in neurons have shown that RHOT1 connects the mitochondria via motor proteins to the axon filaments, thereby regulating mitochondrial movement (Schwarz 2013). Therefore, it is likely that RHOT1 is needed for proper mitochondrial movement also in human islets and β -cells, creating a balance in mitochondrial dynamics and maintaining functional mitochondria, as required to maintain the proper ATP production needed for GSIS. Thus, impaired RHOT1 expression alters mitochondrial dynamics, which impacts the number of functional mitochondria, resulting in reduced ATP production and insulin secretion (Fig. 7d).”*

We have also included a new Fig. 7 with a schematic drawing of RHOT1 function in normal and diseased β -cells (**Fig. 7d**). Page 30, line 1179-1185, Figure 7d legend: *“Based on our results, we suggest that RHOT1 is needed for proper mitochondrial movement in β -cells, creating a balance in mitochondrial dynamics to maintain functional mitochondria. Left: Functional mitochondria with balanced fission, fusion, and mitochondrial movement. Right: Impaired RHOT1 expression may affect mitochondrial movement with consequences for mitochondrial dynamics (unbalanced fusion/fission processes and altered mitophagy rate). A reduced number of functional mitochondria may impact ATP production and ultimately impair GSIS.”*

3. The authors found an almost complete ablation of Rhot1 protein levels in islets of adult GK rats with the decrease of insulin secretion. Therefore, what is the relationship between Rhot1 expression and insulin secretion in GK rats. It would be helpful to analyze the correlation between Rhot1 expression and insulin secretion.

Response: Based on this valid comment, we correlated the levels of Rhot1 in islets with glucose-stimulated insulin secretion in the adult GK rats. Although the Rhot1 levels are very low in the adult GK rats (see Fig. 6g), there is a trend (insignificant) showing a positive correlation between the Rhot1 levels and insulin secretion (see left figure below), indicating that lower Rhot1 levels are linked to decreased insulin secretion. Moreover, in the Wistar rats, where the Rhot1 levels are higher, we found a positive correlation between the levels of Rhot1 in islets and glucose-stimulated insulin secretion ($p=0.035$), further supporting this hypothesis (see right figure below). We added this information on page 10-11, line 388-390 of the revised ms:

“The Rhot1 protein levels in islets correlated positively with GSIS in Wistar rats ($r=0.90$ and $p=0.036$), but the correlation was insignificant in adult GK rats ($r=0.35$, $p=0.65$), in which both Rhot1 and insulin secretion levels are very low.”

4. The authors found the important role of RHOT1 in maintaining mitochondrial function, but there was no data related to the mitochondrial function in GK rats. It would be better to provide the related results to analyze the relationship between Rhot1 expression and mitochondrial function in GK rats.

Response: Thanks for this valid suggestion of investigating mitochondrial function in GK rats. To do this, we performed Western blot analysis of citrate synthase (a mitochondrial matrix TCA cycle enzyme and marker of intact mitochondria) and of components of all five complexes of the electron transport chain using protein available from the same GK and Wistar rats as were analyzed for Rhot1. Here, citrate synthase protein levels were significantly lower in adult GK versus Wistar rats and there was a nominal reduction in adult versus young GK rats (see **Fig. 6l, Supplementary Fig. 7f**). Moreover, components of all five complexes of the electron transport chain were significantly lower in adult GK versus Wistar rats and protein levels of four components were lower in adult versus young GK rats (see **Fig. 6m-q, Supplementary Fig. 7g**).

We added this information on page 11 (Results) and 12 (Discussion) of the revised ms, in the methods on page 22 and in the legend of **Fig. 6** and **Supplementary Fig. 7**:

Page 11, line 391-397: *“Moreover, the protein level of citrate synthase, a mitochondrial matrix TCA cycle enzyme and marker of intact mitochondria, was significantly reduced in islets from adult GK versus Wistar rats, while the reduction in adult versus young GK rats was nominal (Fig. 6l, Supplementary Fig. 7f). We also found reduced protein levels of components from all five complexes in the electron transport chain in islets from adult GK versus Wistar rats, of which four were also reduced in adult versus young GK rats (Fig. 6m-q, Supplementary Fig. 7g). These results are in line with the data in Rhot1-deficient β -cells and support mitochondrial dysfunction in islets from the GK rat.”*

Page 12, line 465-467: *“We also found an age-related decline in Rhot1 and lower levels of several components of the electron transport chain in islets from diabetic GK rats, supporting the role of Rhot1 in islet function and T2D.”*

Page 21, line 802-806: *“For Western blot analysis, rat islets were lysed in RIPA buffer and the analysis performed as described above for INS-1 832/13 β -cells using the RHOT1 primary antibody (ab188029, abcam), cytochrome C antibody (#4272T; 1:1000, Cell Signaling Technologies) and total OXPHOS rodent antibody cocktail including antibodies against NDUFB8, SDHB, UQCRC2, MTCO1, and ATP5A (#ab110413-MS604, 1:250, abcam).”*

Page 30, line 1160-1167: *“(l) Citrate synthase protein levels were lower in islets from adult GK (n=4) vs. Wistar (n=5) male rats (p=0.02) and there was a nominal reduction in adult (n=4) vs. young (n=4) male GK rats (p=0.098) based on t-tests after log-transformation to reach normal distribution. Target signal was normalized to the total amount of protein loaded in each lane. m-q) Reduced protein levels of five and four components of the electron transport chain in islets from adult GK (n=4) vs. control Wistar rats (n=5) as well as in adult vs. young (n=4) GK rats, respectively, based on t-tests after log-*

transformation to reach normal distribution. Target signal was normalized to the total amount of protein loaded in each lane.”

Page 33, line 1269-1274: e-g Western blot showing protein levels of RHOT1 (e), CS (f), and five components of the electron transport chain (NDUFB8, SDHB, UQCRC2, MTCO1, and ATP5A) (g) in adult (12 weeks of age; n=4) vs. young (8 weeks of age; n=4) GK rats and control Wistar rats (12 weeks of age; n=5) (top). Protein levels were normalized to the total amount of protein loaded in each lane (bottom) using ImageLab software version 6.0.1 (V3 Western Workflow, Bio-Rad). siNC: non-target control.

5. In Figure 3, RHOT1 expression was decreased by all four siRNA in human islets, such as FOXP1, TBC1D4, RHOT1 and CABLES1. Therefore, how does FOXP1, TBC1D4 and CABLES1 regulate the expression of RHOT1.

Response: Thanks for this question; it is indeed an interesting finding that some genes are affected by silencing of several other genes. For *RHOT1* downregulation, different mechanisms may be involved, as suggested by information in different databases, FOXP1 has a binding site in the *RHOT1* promoter, whereas TBC1D4 and *RHOT1* may interact on the protein level. This has now been included in the Discussion, Page 12, line 452-454: “*The reduction in RHOT1 may have different causes; for example, FOXP1 is a TF with a binding site in the RHOT1 promoter, whereas TBC1D4 and RHOT1 display a protein-protein interaction (Rouillard, 2016).*”

Based on this valid comment, we also performed a new analysis to dissect how FOXP1, TBC1D4 and CABLES1 potentially may regulate the expression of *RHOT1* and included these results on page 8, line 284-297:

“To characterize potential mechanisms underlying the reduced RHOT1 expression when silencing FOXP1, TBC1D4, and CABLES1 in human islets, we identified 619 TF binding sites in the vicinity of the RHOT1 gene (-1000, 0) using publicly available ChIP-seq data from the Gene Transcription Regulation Database (GTRD)(Kolmykov, Yevshin et al. 2021) (Supplementary Fig. 5e). Since most experiments in the GTRD were performed in non-islet cell types, we examined whether the investigated genomic regions are accessible in human islets, based on published ATAC-seq and histone modification data(Miguel-Escalada, Bonas-Guarch et al. 2019). We identified 362 TFs with 614 binding sites within regions defined as promoters in pancreatic islets (Supplementary Table 16). Next, we searched for overlaps between the 362 TFs that could bind 0-1000 bp upstream of RHOT1 TSS and differentially expressed genes after silencing FOXP1, CABLES1, or TBC1D4 in human islets (Supplementary Table 15, 16). We detected 7, 8 and 13 differentially expressed TFs, and among these, 6, 4 and 8 were downregulated after silencing FOXP1, CABLES1 or TBC1D4, respectively (Supplementary Table 16, Supplementary Fig. 5e). Although dysregulation of each of these TFs could contribute to the reduction of RHOT1, one TF, MLXIP (formerly known as MONDOA), was found to be dysregulated in all 3 silencing experiments. MLXIP is an important glucose-sensing TF in human β -cells(Richards, Rachdi et al. 2018).”

6. In figure 4c and 5g, there was no GAPDH or ACTIN. How does the authors perform the quantification in WB analysis. Please explain that in detail.

Response: We are sorry that the description of the method for Western blot normalization was not fully clear. We have now further explained how this was done, including a reference, in the method section:

Page 17, line 658-663: “*The antibody-specific signal was quantified and we then used total protein normalization where the detected antibody-specific signal is normalized to total protein in ImageLab*

software version 6.0.1 (V3 Western Workflow, Bio-Rad). With total protein normalization, the antibody specific signal is normalized to the total amount of protein loaded in each lane. This method is more stable for different loading amounts than using house-keeping genes and avoids the risk of the internal control being affected by experimental conditions (Gurtler et al., 2013)."

We also added more detailed info into the figure legends: page 29-30: "**5c** *RHOT1 protein levels in Rhot1-deficient β -cells with the target signal normalized to the total amount of protein in each lane*" and "**6g** *Rhot1 protein in 8- and 12-week-old (young and adult, respectively; n=4) GK rats and 12-week-old control Wistar rat (n=5) with the target signal normalized to the total amount of protein in each lane*"

For consistency, we also changed the legends for supplementary figures accordingly (Page 32, Figure legend for **Supplementary Fig. 5h, 6a-c, and 7a-g**).

7. In some Figure legends, the sample size n was missed.

Response: Thanks for this thorough review. We have now gone through all figure legends to ensure that sample sizes are included. Also, we also updated the figure legends to meet the Nature Communications formatting guidelines. See the updated **Figure legends**, page 28-33.

Reviewer #3 (Remarks to the Author):

The manuscript presents a genome-wide DNA methylation study comparing healthy (75 samples) and T2D (25 samples) human islets. The authors also examine the correlation between differential DNA methylation and HbA1c using islets from an independent cohort (114 samples). They integrate the methylation data with RNA-seq, ATAC-seq, transcription factor and histone modifications' ChIP-seq data. They reported four T2D-associated methylation sites associated with lower risk of T2D in blood samples. Selected candidate genes are validated using siRNA knockdown in a rat beta cell line. The study highlights the potential role of Rhot1 in regulating mitochondria function and metabolites, as well as its deficiency in a rodent diabetes model associated with reduced insulin content. Although this study has multiple points, the logical flow between them is not coherent or compelling. Although the authors hint a mechanistic connection between DNA methylation and gene expression, the evidence is preliminary. A lot of results are only described in the text with a brief reference to an excel file without proper presentation or quantitative analysis. A key result about the prospective T2D risk analysis not adequately presented or discussed. Overall, I feel that this manuscript has major flaws and at least needs a lot of improvements to be acceptable for publication in NC. Here are some specific points.

Response: We appreciate the valuable comments provided by Reviewer #3, which helped us improve our study and manuscript considerably. We have addressed all review comments, both from the above section, and the specific points below. We believe the revised manuscript and our novel data are of importance and of immediate interest to the broad readership of Nature Communications.

Response to the more general comments above:

"Although this study has multiple points, the logical flow between them is not coherent or compelling"

Response: We want to thank Reviewer 3 for this comment. To make research findings easily available for the scientific community, the flow within the manuscript is indeed very important. To improve readability and to make the message and results clearer, we worked together with a professional proof-reading and editing company (San Francisco Edit), to optimize the manuscript before re-submission.

We believe this editing substantially improved the manuscript and answered this as well as other concerns. Please see changes marked with track-changes throughout the re-submitted manuscript.

“Although the authors hint a mechanistic connection between DNA methylation and gene expression, the evidence is preliminary”

Response: We appreciate this comment and, in our opinion, the connection between DNA methylation and gene expression is well established within the epigenetic research community. For example, we have several times shown functionally that increased DNA methylation within gene promoters directly regulates (reduce) the transcriptional activity and we already referred to some of these studies in our Results, page 5, line 157-158: *“DNA methylation may interfere with transcriptional activity, as proven by in vitro experiments^{4,6} and causal inference test¹⁷”*. This notwithstanding, to again prove a mechanism for epigenetic gene regulation, we designed a Luciferase assay by cloning the *RHOT1* promoter region into a vector lacking CpG sites. By methylating the promoter sequence *in vitro*, we were able to downregulate the transcriptional activity in clonal β -cells. This data is now added on page 8-9, line 305-310, and in the methods on page 16, line 617-627, and in **Supplementary Fig. 5f**.

“First, we investigated the direct impact of increased promoter DNA methylation on the transcriptional activity of RHOT1, using a luciferase assay (Supplementary Methods). The CpG-free luciferase reporter construct including 1540 bp of the RHOT1 promoter was methylated by SssI and transfected into INS-1 832/13 β -cells. Methylation of the promoter reduced the transcriptional activity (Supplementary Fig. 5f), supporting that altered DNA methylation can influence gene expression.”

“Luciferase assay

1540 bp of the RHOT1 promoter (Supplementary Methods) was inserted upstream of TSS of a CpG-free luciferase reporter vector (pCpGL-basic, kindly provided by Dr Klug and Dr Rehli⁷⁴). SssI, a DNA methyltransferase (New England Biolabs, Ipswich, MA, USA), was then used for methylation of the construct. INS-1 832/13 β -cells were co-transfected with 25 ng methylated pCpGL-vector including the RHOT1 promoter together with 4 ng of pRL renilla luciferase control reporter vector (Promega, Madison, WI, USA). Firefly luciferase luminescence, as the value of transcriptional activity, was measured using the Dual-Luciferase® Reporter Assay System (Promega) and an Infinite® M200 PRO multiplate reader (Tecan Group Ltd., Männedorf, Switzerland). Cells transfected with an empty pCpGL-vector were used as background control for firefly luciferase results, and untransfected cells were used as a background for renilla results.”

“A lot of results are only described in the text with a brief reference to an excel file without proper presentation or quantitative analysis”

Response: We agree that some results were only briefly presented in the running text, however as you mentioned, all data were to be found in figures and/or supplementary tables. This has now been improved, partly by the use of professional editing as specified above, and partly when doing the final editing to meet journal requirements. Please see the re-submitted version of our manuscript.

“A key result about the prospective T2D risk analysis not adequately presented or discussed”

Response: We are sorry that this section was not adequately presented. We have now expanded on these results, both in the Results and the Discussion. For details, please see our response to comment 6 below.

Point-by-point response to specific comments:

1. I highly recommend including a panel in figure 1 (such as a heatmap) showing the changes the DNA methylation changes in every individual. This will help readers understand what kind of changes we are looking at, and how consistent the changes are between the

control and T2D groups. Without a global picture, the readers may feel that the authors are only cherry-picking some best examples.

Response: Thanks for this comment. We agree that heatmaps can be a good way to visualize global changes in large datasets. However, you can only visualize a certain number of datapoints within one figure, within the magnitude of a few thousands, but to include and properly visualize ~850,000 rows is not possible. Doing that would require to collapse several methylation sites into one data point, thereby losing the intention of studying a large amount of methylation sites with possible specific effects, rather than just measure global, overall methylation.

Nevertheless, to show the consistency between individuals from the T2D cases and control groups, we have now included a Heatmap in **Fig. 1**, showing DNA methylation levels for each individual, for the 100 CpG sites with the highest absolute differences in methylation between groups (Results, Page 3, line 86-88; Fig. 1e; and page 28, line 1088-1090 Figure legend 1e). *“A heatmap of the 100 sites with largest differences in methylation between T2D cases and controls visualizes the consistency between individuals in each group (Fig. 1e)”* and *“Hierarchical clustering heatmap showing the DNA methylation pattern for the 100 CpG sites with the largest differences in the Islet T2D case-control cohort. Samples are shown on the x-axis, and methylation sites on the y-axis.”*

Also, **Fig. 2a-d** clearly show that for specific sites, methylation levels are consistent between the T2D cases and control groups. Moreover, Supplementary Table 1, 3, and 4 present all our epigenome-wide associations and in these tables, we present the regression coefficients together with the standard error as well as mean and SD for respective group and differences between the groups, both absolute differences and fold-change. We believe we are transparent with all our data/results, and our goal was to avoid cherry picking. To generate epigenome-wide associations, we used robust statistical analyses, correction for multiple testing and correction for cell composition as presented in the methods.

2. The p-values for the enrichment in Figure 1e is missing. The significance of the enrichment is not clear.

Response: Thanks for letting us know that this needs to be better explained. We have now updated the legend for Figure 1e (now **Fig. 1f**), page 28, line 1090-1093:

“f) Selected KEGG pathways associated with T2D based on genome-wide DNA methylation data from the Islet of T2D case-control cohort (dark blue; adj. P-values <0.05). Most pathways were also associated with HbA1c, based on genome-wide DNA methylation data from the Islet HbA1c cohort (light blue; adj. P-values <0.05).”

Additionally, we have added all individual adjusted p-values into **Fig. 1f** (All unique p-values were previously only found in **Table S2**).

3. The authors claimed some consistency between current study and previous studies (Dayeh et al.; Volkov et al.) but provide very little details. Only an excel file is provided. Summarization of the comparison should be included. For example, how many hyper-methylated sites from this study are hyper or hypo- methylated in the other studies; conversely, how many hypo-methylated sites from this study are hyper or hypo- methylated in the other studies. Quantitative statistical analysis should be also included.

Response: We appreciate this comment and have included the requested information on page 4, line 121-128 of the revised ms:

“We also compared the current EPIC results with a previous study using 450K arrays and islets of 15 T2D cases and 34 controls(Dayeh, Volkov et al. 2014). There was a small overlap of islet donors with the current study (5 cases, 18 controls). Importantly, of sites reported by Dayeh et al. to have altered methylation in islets from T2D cases ($q < 0.05$ and $\Delta\beta \geq 5\%$), we could replicate 813 sites with consistent differences between cases and controls (Supplementary Table 7), including sites annotated to CACNA1C, CDKN1A, GLP1R, HDAC4, HDAC7, IL6R, KCNQ1, PDE7B, and THADA (Supplementary Fig. 1a-i). In both studies, 98.5% of these 813 sites were hypomethylated in islets from T2D cases versus controls (Supplementary Table 7).”

Importantly, here, we only consider results consistent when the different studies show changes in the same direction, e.g., in the comparison with Dayeh et al., sites are hypomethylated both in that study and in the current EPIC T2D case-control cohort, or sites are hypermethylated in both studies. This has now been highlighted in the manuscript and is also clearly shown in Supplementary Table 7.

Regarding statistical enrichment analysis between the current study and the study by Dayeh et al.; we do not believe it is correct to do such an analysis due to the overlap of some samples between the different studies, meaning that the different cohorts are not fully independent. Hence, we present numbers (within the text) and specific sites (Supplementary Table 7), to highlight that these sites/genes have been significantly associated with T2D repeatedly. Moreover, the following sentence was included on page 4: “There was a small overlap of islet donors with the current study (5 cases, 18 controls).” It should be noted, however, that a Chi²-test found the overlap enriched ($p < 0.05$, not included in the revised ms).

While it is possible to include all the requested information for the study by Dayeh et al., which investigated DNA methylation of individual CpG sites using the 450k Illumina array, it is not possible to do the same for the study by Volkov et al., which studied differentially methylated regions (DMRs) using whole genome bisulfite sequencing (WGBS). For DMRs, methylation differences presented are based on the average of multiple sites, as described on page 4, line 129-136. Nevertheless, some of the requested information was added here.

Next, we compared our data to a previous whole genome bisulfite sequencing (WGBS) analysis of DNA methylation in islets from six T2D cases and eight controls (Volkov, Bacos et al. 2017). We found that 1,297 of the methylation sites associated with T2D and 712 of the sites associated with HbA1c in the current study are within differentially methylated regions (DMRs) identified by WGBS (**Supplementary Table 7**). DMRs contain consecutive sites (range 3-164) and reported methylation differences are averages for the whole DMR ($\geq 5\%$). These include sites annotated to DNMT3A, GLP1R, HDAC4, IL6R, KCNQ1, PDX1, TCF7L2, THADA, and WFS1 (**Supplementary Fig. 1c-d, f-g, i-m**). *72% of the 1,297 sites were hypomethylated in islets from T2D cases versus controls (Supplementary Table 7).*”

4. The changes of DNA methylation shown in Fig. 2 examples all very subtle, making me wonder if these changes are meaningful. Figures 2a-2d and S2a-2c would benefit from being shown in a genome browser with larger range, with the information of other CpG sites and the results of nearby genes.

Response: Thanks for this comment. Indeed, some of the significant methylation differences are subtle, but importantly, they are plenty. It is commonly known that in multifactorial, polygenic diseases, such as type 2 diabetes, many smaller changes have additive effects and together increase risk of disease. In support of this hypothesis and to see the overarching effect, we created combined methylation risk scores (MRSs) for genes with altered expression and more than 5 differentially methylated CpG sites within or nearby (± 10 kb) that gene (in total 27 genes, with differential methylation of 6-30 CpG sites per gene). We also tested if the MRSs were associated with higher risk of having type 2 diabetes based on odds ratios (OR). All genes displayed significant MRSs, and all MRSs were associated with higher risk of T2D, supporting combined effects of multiple CpG sites on the risk of T2D. These data are now included in the Result section Page 5, line 175-186, and in Supplementary Fig. 4 and Supplementary Table 10:

*“Next, we calculated a weighted combined methylation risk score (MRS) for each gene with altered expression and more than five differentially methylated CpG sites within or near (± 10 kb) that gene, to study the combined effect of multiple differentially methylated sites in individual genes (total 27 genes; **Supplementary Table 10**). All MRSs were different between T2D cases versus controls in multiple linear regression models ($p < 1.1 \times 10^{-4}$). MRSs for SYNPO ($\beta = 84.6 \pm 11.1$, $p = 2.1 \times 10^{-11}$) and FOXP1 ($\beta = 57.7 \pm 8.1$, $p = 2.2 \times 10^{-10}$) had the biggest effect size in the linear models, displaying that individuals with T2D have a higher MRS compared to controls. **Supplementary Fig. 4a-f** presents the MRS for BEST3, CABLES1, FAIM2, FOXP1, SLC2A2 and TBC1D4, demonstrating the combined effect of several differentially methylated sites on T2D. Moreover, when performing multiple logistic models, higher values of all MRS were associated with a higher risk of having T2D with odds ratios (OR) ranging between 1.03 and 2.66 per 1% increase in methylation ($p < 5.4 \times 10^{-4}$; **Supplementary Table 10** and **Supplementary Fig. 4g**).”*

The statistics behind the MRS calculations are included in the method section, Page 21, Line 825-832:

“To integrate epigenetic information across different methylation sites, we calculated a weighted combined MRS for each gene including genes with altered expression and more than 5 differentially methylated CpG sites within or near (± 10 kb) that gene (27 in total). These weighted MRSs were calculated as the sum of the methylation values (%) at each included CpG site in the gene, weighted by CpG-specific effect size (multiplying by β -coefficients from the multiple linear regressions explained above when assessing differences in methylation between T2D cases compared with controls) (Schrader, Perfilyev et al. 2022). Then, multiple linear and logistic regression models adjusted for age, sex, BMI, islet purity and days in culture were performed to assess the association between MRSs and T2D.”

Moreover, on page 3, line 78-86, and in **Fig. 1b-d**, we present the largest differences in DNA methylation of individual CpG sites:

“The largest absolute methylation difference was 21.1%, at cg15132295 in DNAH17 (Fig. 1b). Lower DNAH17 methylation has been linked to hepatic cancer(Fan, Guo et al. 2019). For increased methylation in T2D cases, the largest absolute difference was 19.5% (cg09972436 in LCE3C, Fig. 1c), a gene linked to autoimmune diseases (Docampo, Rabionet et al. 2010). We also calculated the fold change in DNA methylation, as the impact of absolute differences may depend on whether basal methylation levels are low or high. We found percentage differences up to 80%, meaning that the number of cells being methylated at a specific site was almost doubled in one group (Table S1). This was true for a site in DSCR3, with 12.9% methylation in controls and 23.3% methylation in T2D cases (Fig. 1d). DSCR3 methylation is a potential biomarker for preeclampsia (Kim, Kim et al. 2015).”

And on page 3-4 we present pathway analyses, further pointing to enrichment of pathways linking islet DNA methylation to risk of T2D:

Line 89-94: “To further explore this comprehensive data and its biological context, we used the methylGSA package to perform a pathway analysis on the whole methylation data set (Ren and Kuan 2019). Seventy-two KEGG pathways were associated with T2D (adj. p-value <0.05), including the MAPK, Calcium, and Insulin signaling pathways, Type II diabetes mellitus, Pancreatic secretion, and Purine metabolism (Supplementary Table 2). Fig. 1f visualizes some key pathways, supporting a strong link between islet methylation and T2D.”

Line 108-112: “A pathway analysis on the whole methylation data set based on HbA1c associations revealed 53 enriched pathways (adj. p-value <0.05). The result resembled that of T2D associations, and 47 pathways overlap with pathways enriched in T2D, including Type II diabetes mellitus and Pancreatic secretion (Supplementary Table 5, Fig. 1f).”

5. It is unclear whether the differential methylation sites are only enriched in the DEG genes and how significant this is.

Response: We appreciate this question. However, we do not believe it is appropriate to calculate significance for enrichment of differential methylation in specific genes in a study based on the EPIC array. The array covers ~850,000 CpG sites, however, these are *not* evenly distributed throughout the genome. Different genes are represented by a highly variable number of CpG sites, and it also differs where in the gene those CpG sites are located. Moreover, the array is based on 50 bp probes and single-base extension, which further limits the possibility to cover all genomic features. The overlap of DEG and altered methylation in our study varies between 1-30 significant CpG sites, which could be weighted in different ways (e.g. gene location, number of CpGs in the region, presence of CpG island etc). Hence, we present the overlap in the manuscript “75% of genes with differential expression in islets of T2D cases versus controls also display one or more sites with differential methylation”, however, we do not argue that this is a significant enrichment, nor do we believe this is an appropriate test to do with this study design.

Anyway, we performed a Chi²-test as this reviewer specifically asked for significance, and indeed, there is an enrichment of differentially methylated sites in DEGs (p=2x10⁻²¹).

Chi ² -test of protein coding genes that are differentially expressed in T2D cases versus controls (DEG; columns), and also displaying differential methylation in T2D cases versus controls (rows)			
OBSERVED	DEG	Other	all
Singificant	153	8329	8482
Non-significant	50	11369	11419
EXPECTED	DEG	Other	all
Singificant	86,52	8395,48	8482
Non-significant	116,48	11302,52	11419
All genes	203	19698	19901
chi-square test		2,45447E-21	

DEG: Differentially expressed gene, with one or more differentially methylated CpG sites within +/- 10 kb

We will leave it to the reviewer and editor to decide if we should include the significance of enrichment, or if it is misleading, based on the discussion above.

6. The section "DNA methylation associates with future T2D" is extremely brief. Again, the authors only refer the readers to an excel file. Here the authors test the putative CpG sites in blood samples. Should we worry about the consistency between islets and blood? This

should be at least discussed. In fact, the authors could have screened for candidate predictive DNA methylation sites from blood samples in the beginning. Wouldn't that give better outcomes? The authors names 4 predictive sites, including one with RHOT1, with low T2D risk and low DNA methylation. But in Fig. 1, the RHOT1 CpG show low DNA methylation among T2D patients (i.e., low DNA methylation is associated with increased T2D risk). This is contradictory.

Response: We agree with reviewer 3 that this section is brief and based on this comment we added more information in the result section on page 7, in Supplementary Table 14 and in the discussion on page 11-12.

Regarding consistency between the epigenetic pattern in pancreatic islets and blood, the epigenome has a main function in regulating cell specific gene expression and shows large differences between different cell types (*Roadmap Epigenomics Consortium, Nature, 2015, PMID: 25693563*). Nevertheless, certain CpG sites can have the same epigenetic pattern and regulation under certain conditions in several different cell types. For example, we previously demonstrated that ageing is associated with alterations in DNA methylation of the same sites/genes (e.g., *ELOVL2, FHL2, KLF14*) in several different cell types such as human adipose tissue, blood, pancreatic islets and the liver (*Rönn et al Hum Mol Gen, 2015, PMID: 25861810, Bacos et al Nature Com, 2016, PMID: 27029739, and Bysani et al Epigenomics, 2017, PMID: 27911095*), hence supporting that methylation of some sites in blood may mirror the methylation pattern in pancreatic islets.

The overall goal of this study was to better understand the role of epigenic dysregulation in the pathogenesis of T2D, and since pancreatic islets is a key tissue influencing the development of diabetes due to its secretion of insulin, we chose to study DNA methylation in human pancreatic islets. Blood has a less important role in the pathogenesis of T2D, and therefore blood was not the focus of this study. However, if one can identify some sites with differential DNA methylation in pancreatic islets from donors with T2D also showing similar regulation in blood, this may be used as prognostic biomarker. This is the reason for why we also analyzed DNA methylation in blood of a prospective cohort of CpG sites with differential methylation in islets from donors with T2D. It should also be noted that some studies have previously analyzed DNA methylation in blood using either the Infinium 450k array or the EPIC array, and together these studies only found a few methylation sites associated with T2D (ref 27-33 in the ms). Hence, we do not believe screening for candidate predictive DNA methylation sites from blood samples in the beginning would have been better.

However, based on this valid comment, we performed a search of previous T2D DNA methylation studies carried out in blood, summarized these results, and then checked for overlap with the methylation sites identified in islets from the present study.

Page 7, line 240-247: “*We next compared our identified sites showing differential DNA methylation in pancreatic islets from T2D cases versus controls (Supplementary Table 1), with published DNA methylation data from prospective or incident T2D studies performed in blood (Chambers, Loh et al. 2015, Florath, Butterbach et al. 2016, Cardona, Day et al. 2019, Juvinao-Quintero, Marioni et al. 2021, Domingo-Relloso, Gribble et al. 2022, Fraszczyk, Spijkerman et al. 2022, Fraszczyk, Thio et al. 2022). Previous studies in blood identified two methylation sites (cg00574958/CPT1 and cg02711608/SLC1A5) also identified here in human islets (Supplementary Table 14) (Cardona, Day et al. 2019, Juvinao-Quintero, Marioni et al. 2021, Fraszczyk, Spijkerman et al. 2022). Moreover, previous T2D studies performed in blood found differential methylation of the same genes as we identified in the Islet T2D case-control cohort (e.g., TXNIP, ABCG1, and HDAC4), but the specific sites identified in blood were different from the methylation sites identified in islets (Supplementary Table 1, 14) (Chambers, Loh et al. 2015, Florath, Butterbach et al. 2016, Cardona, Day et al. 2019, Juvinao-Quintero, Marioni et al. 2021, Domingo-Relloso, Gribble et al. 2022, Fraszczyk, Spijkerman et al. 2022, Fraszczyk, Thio et al. 2022).”*

Page 11-12, line 428-435: “Moreover, a previous study found 18 methylation sites in blood associated with future T2D (Cardona, Day et al. 2019). Of these, two were among our T2D-associated methylation sites in human islets (cg00574958/CPT1A and cg02711608/SLC1A5), and the first one was also associated with HbA1c. Two additional studies identified T2D-associated methylation of cg00574958/CPT1A in blood (Juvinao-Quintero, Marioni et al. 2021, Fraszczyk, Spijkerman et al. 2022). Other studies performed in blood identified methylation sites associated with T2D, and several of these sites were annotated to the same genes, but for different CpG sites, as were differentially methylated in human islets of T2D donors in the present study (Chambers, Loh et al. 2015, Florath, Butterbach et al. 2016, Cardona, Day et al. 2019, Juvinao-Quintero, Marioni et al. 2021, Domingo-Reloso, Gribble et al. 2022, Fraszczyk, Spijkerman et al. 2022, Fraszczyk, Thio et al. 2022). Epigenetic alterations common for several tissues may provide therapeutic or predictive biomarkers.”

Finally, regarding the degree of methylation of the four predictive sites identified in our study: Odds ratio <1 means lower T2D risk with increasing DNA methylation (as seen in the EPIC-Potsdam), which is in accordance with lower DNA methylation in the islet T2D cases. To clarify the different ways to interpret the outcomes, depending on type of analysis, we have now included results for both linear and logistic regression for the Islet T2D case-control cohort in Table 2, page 27: “For islet DNA methylation, both absolute difference (cases-controls) based on linear regressions and odds ratios based on logistic regressions are shown.” And as can be seen, the results in blood and pancreatic islets are in agreement, although two separate tissues:

“Table 2. DNA methylation associated with T2D in the Islet T2D case-control cohort within genes differentially expressed in islets from T2D cases vs. controls that also associated with future T2D based on blood DNA methylation in the prospective EPIC-Potsdam T2D case-control study.

CpG site	Islet T2D case-control cohort				EPIC-Potsdam			
	DNA methylation				RNA-Seq		Blood DNA methylation	
	Absolute difference	P-value	Odds ratio (95% CI)	P-value	Gene	P-value	Odds ratio (95% CI)	P-value
cg05324789	-3.50%	3.2x10 ⁻⁴	0.77 (0.65-0.90)	0.0014	NKX6.2	4.3x10 ⁻⁵	0.43 (0.24-0.76)	0.004
cg20355381	-5.60%	4.4x10 ⁻⁷	0.63 (0.47-0.78)	3.1x10 ⁻⁴	CABLES1	1.2x10 ⁻⁴	0.56 (0.37-0.84)	0.006
cg15167202	-4.70%	6.8x10 ⁻⁴	0.85 (0.76-0.94)	0.0027	SYNPO	3.0x10 ⁻⁴	0.43 (0.20-0.94)	0.034
cg10339923	-8.30%	5.0x10 ⁻⁸	0.76 (0.66-0.86)	4.9x10 ⁻⁵	RHOT1	3.3 x10 ⁻⁴	0.68 (0.47-0.97)	0.035

For islet DNA methylation, both absolute difference (cases-controls) based on linear regressions and odds ratios based on logistic regressions are shown.”

7. Fig. 5g-k shows that in diabetic rat, the down regulation of Rhot1 is correlated with lower insulin content in the islets, suggesting a causal role. However, in human islets (Fig. S4a), siRHOT1 does not affect islet insulin content. This raise a serious question if and how much of the Rhot1 functional results from rat are conserved in human islets.

Response: We thank Reviewer 3 for this question. Human islet experiments were performed in islets from previously healthy/non-diabetic donors, where RHOT1 is being silenced *in vitro* for 72h, and where insulin secretion/insulin content was measured and shown to be reduced. Similarly, there is a reduction in GSIS in the GK-rat islets when comparing insulin secretion/insulin content (2.8 ± 0.7 vs. 5.2 ± 0.7, p=0.007). Hence, RHOT1 impairs the insulin secretion process independently of changes in insulin content in both human islets *in vitro* and in islets from the GK rat. It is true that insulin content is reduced in the GK-rat islets, but Rhot1 is probably one of several factors, which together contribute to diabetes and/or reduced insulin content in the GK rats (see e.g.: Nagao, M. et al. 2020. Selectively Bred Diabetes Models: GK Rats, NSY Mice, and ON Mice. In: King, A. (eds) Animal Models of Diabetes. Methods in Molecular Biology, vol 2128. Humana, New York, NY. https://doi.org/10.1007/978-1-0716-0385-7_3). Moreover, we previously found reduced insulin content in human pancreatic islets from

donors with T2D versus non-diabetic controls (see figure below, *BT Yang et al Diabetologia 2012, PMID: 21104225*).

[REDACTED]
Figure 1b-c, Yang et al

Based on this comment, we added the following information to the Discussion on page 13, line 484-491: *“While GK rats showed a decline in Rhot1 and OXPHOS protein levels as well as in insulin secretion and content, silencing of RHOT1 for 72 h in human islets from non-diabetic donors reduced GSIS and caused mitochondrial dysfunction without changing the insulin content. The discrepancy in insulin content between the two models may be due to that the knock-down specifically focus on the effect of RHOT1, whereas in the GK rat, Rhot1 is probably one of several factors, which together contribute to diabetes. Moreover, we previously found reduced insulin secretion and content when comparing islets from human donors with T2D versus non-diabetic controls (Yang et al., Diabetologia 2012), and in the present study we found reduced Rhot1 in the same setting.”*

References:

Cardona, A., F. R. Day, J. R. B. Perry, M. Loh, A. Y. Chu, B. Lehne, D. S. Paul, L. A. Lotta, I. D. Stewart, N. D. Kerrison, R. A. Scott, K. T. Khaw, N. G. Forouhi, C. Langenberg, C. Liu, M. M. Mendelson, D. Levy, S. Beck, R. D. Leslie, J. Dupuis, J. B. Meigs, J. S. Kooner, J. Pihlajamaki, A. Vaag, A. Perflyev, C. Ling, M. F. Hivert, J. C. Chambers, N. J. Wareham and K. K. Ong (2019). "Epigenome-Wide Association Study of Incident Type 2 Diabetes in a British Population: EPIC-Norfolk Study." *Diabetes* **68**(12): 2315-2326.

Chambers, J. C., M. Loh, B. Lehne, A. Drong, J. Kriebel, V. Motta, S. Wahl, H. R. Elliott, F. Rota, W. R. Scott, W. Zhang, S. T. Tan, G. Campanella, M. Chadeau-Hyam, L. Yengo, R. C. Richmond, M. Adamowicz-Brice, U. Afzal, K. Bozaoglu, Z. Y. Mok, H. K. Ng, F. Pattou, H. Prokisch, M. A. Rozario, L. Tarantini, J. Abbott, M. Ala-Korpela, B. Albeti, O. Ammerpohl, P. A. Bertazzi, C. Blancher, R. Caiazzo, J. Danesh, T. R. Gaunt, S. de Lusignan, C. Gieger, T. Illig, S. Jha, S. Jones, J. Jowett, A. J. Kangas, A. Kasturiratne, N. Kato, N. Kotea, S. Kowlessur, J. Pitkaniemi, P. Punjabi, D. Saleheen, C. Schafmayer, P. Soinenen, E. S. Tai, B. Thorand, J. Tuomilehto, A. R. Wickremasinghe, S. A. Kyrtopoulos, T. J. Aitman, C. Herder, J. Hampe, S. Cauchi, C. L. Relton, P. Froguel, R. Soong, P. Vineis, M. R. Jarvelin, J. Scott, H. Grallert, V. Bollati, P. Elliott, M. I. McCarthy and J. S. Kooner (2015). "Epigenome-wide association of DNA methylation markers in peripheral blood from Indian Asians and Europeans with incident type 2 diabetes: a nested case-control study." *Lancet Diabetes Endocrinol* **3**(7): 526-534.

Dayeh, T., P. Volkov, S. Salo, E. Hall, E. Nilsson, A. H. Olsson, C. L. Kirkpatrick, C. B. Wollheim, L. Eliasson, T. Ronn, K. Bacos and C. Ling (2014). "Genome-wide DNA methylation analysis of human pancreatic islets from type 2 diabetic and non-diabetic donors identifies candidate genes that influence insulin secretion." *PLoS Genet* **10**(3): e1004160.

Docampo, E., R. Rabionet, E. Riveira-Munoz, G. Escaramis, A. Julia, S. Marsal, J. E. Martin, M. A. Gonzalez-Gay, A. Balsa, E. Raya, J. Martin and X. Estivill (2010). "Deletion of the late cornified envelope genes, LCE3C and LCE3B, is associated with rheumatoid arthritis." *Arthritis Rheum* **62**(5): 1246-1251.

Domingo-Relloso, A., M. O. Gribble, A. L. Riffo-Campos, K. Haack, S. A. Cole, M. Tellez-Plaza, J. G. Umans, A. M. Fretts, Y. Zhang, M. D. Fallin, A. Navas-Acien and T. M. Everson (2022). "Epigenetics of type 2 diabetes and diabetes-related outcomes in the Strong Heart Study." *Clin Epigenetics* **14**(1): 177.

Fan, X., H. Guo, B. Dai, L. He, D. Zhou and H. Lin (2019). "The association between methylation patterns of DNAH17 and clinicopathological factors in hepatocellular carcinoma." *Cancer Med* **8**(1): 337-350.

Florath, I., K. Butterbach, J. Heiss, M. Bewerunge-Hudler, Y. Zhang, B. Schottker and H. Brenner (2016). "Type 2 diabetes and leucocyte DNA methylation: an epigenome-wide association study in over 1,500 older adults." *Diabetologia* **59**(1): 130-138.

Fraszczyk, E., A. M. W. Spijkerman, Y. Zhang, S. Brandmaier, F. R. Day, L. Zhou, P. Wackers, M. E. T. Dolle, V. W. Bloks, X. Gao, C. Gieger, J. Kooner, J. Kriebel, H. S. J. Picavet, W. Rathmann, B. Schottker, M. Loh, W. M. M. Verschuren, J. V. van Vliet-Ostaptchouk, N. J. Wareham, J. C. Chambers, K. K. Ong, H. Grallert, H. Brenner, M. Luijten and H. Snieder (2022). "Epigenome-wide association study of incident type 2 diabetes: a meta-analysis of five prospective European cohorts." *Diabetologia* **65**(5): 763-776.

Fraszczyk, E., C. H. L. Thio, P. Wackers, M. E. T. Dolle, V. W. Bloks, H. Hodemaekers, H. S. Picavet, M. Stynenbosch, W. M. M. Verschuren, H. Snieder, A. M. W. Spijkerman and M. Luijten (2022). "DNA methylation trajectories and accelerated epigenetic aging in incident type 2 diabetes." *Geroscience* **44**(6): 2671-2684.

Juvinao-Quintero, D. L., R. E. Marioni, C. Ochoa-Rosales, T. C. Russ, I. J. Deary, J. B. J. van Meurs, T. Voortman, M. F. Hivert, G. C. Sharp, C. L. Relton and H. R. Elliott (2021). "DNA methylation of blood cells is associated with prevalent type 2 diabetes in a meta-analysis of four European cohorts." *Clin Epigenetics* **13**(1): 40.

Kim, H. J., S. Y. Kim, J. H. Lim, D. W. Kwak, S. Y. Park and H. M. Ryu (2015). "Quantification and Application of Potential Epigenetic Markers in Maternal Plasma of Pregnancies with Hypertensive Disorders." *Int J Mol Sci* **16**(12): 29875-29888.

Kolmykov, S., I. Yevshin, M. Kulyashov, R. Sharipov, Y. Kondrakhin, V. J. Makeev, I. V. Kulakovskiy, A. Kel and F. Kolpakov (2021). "GTRD: an integrated view of transcription regulation." *Nucleic Acids Res* **49**(D1): D104-D111.

Ling, C. and T. Rönn (2019). "Epigenetics in Human Obesity and Type 2 Diabetes." *Cell Metabolism* **29**(5): 1028-1044.

Miguel-Escalada, I., S. Bonas-Guarch, I. Cebola, J. Ponsa-Cobas, J. Mendieta-Esteban, G. Atla, B. M. Javierre, D. M. Y. Rolando, I. Farabella, C. C. Morgan, J. Garcia-Hurtado, A. Beucher, I. Moran, L. Pasquali, M. Ramos-Rodriguez, E. V. R. Appel, A. Linneberg, A. P. Gjesing, D. R. Witte, O. Pedersen, N. Grarup, P. Ravassard, D. Torrents, J. M. Mercader, L. Piemonti, T. Berney, E. J. P. de Koning, J. Kerr-Conte, F. Pattou, I. O. Fedko, L. Groop, I. Prokopenko, T. Hansen, M. A. Marti-Renom, P. Fraser and J. Ferrer (2019). "Human pancreatic islet three-dimensional chromatin architecture provides insights into the genetics of type 2 diabetes." *Nat Genet*.

Ren, X. and P. F. Kuan (2019). "methylGSA: a Bioconductor package and Shiny app for DNA methylation data length bias adjustment in gene set testing." *Bioinformatics* **35**(11): 1958-1959.

Richards, P., L. Rachdi, M. Oshima, P. Marchetti, M. Bugliani, M. Armanet, C. Postic, S. Guilmeau and R. Scharfmann (2018). "MondoA Is an Essential Glucose-Responsive Transcription Factor in Human Pancreatic beta-Cells." *Diabetes* **67**(3): 461-472.

Schrader, S., A. Perflyev, E. Ahlqvist, L. Groop, A. Vaag, M. Martinell, S. Garcia-Calzon and C. Ling (2022). "Novel Subgroups of Type 2 Diabetes Display Different Epigenetic Patterns That Associate With Future Diabetic Complications." *Diabetes Care* **45**(7): 1621-1630.

Schwarz, T. L. (2013). "Mitochondrial trafficking in neurons." *Cold Spring Harb Perspect Biol* **5**(6).

Volkov, P., K. Bacos, J. K. Ofori, J. L. Esguerra, L. Eliasson, T. Ronn and C. Ling (2017). "Whole-Genome Bisulfite Sequencing of Human Pancreatic Islets Reveals Novel Differentially Methylated Regions in Type 2 Diabetes Pathogenesis." *Diabetes* **66**(4): 1074-1085.

REVIEWER COMMENTS

Reviewer #2 (Remarks to the Author):

Authors have addressed my concerns.

Reviewer #3 (Remarks to the Author):

I appreciate that the authors' efforts addressing my questions. However, I still have two major issues regarding the two newly added figures.

1. New Supp Fig. 5f. The authors shows that methylated RHOT1 promoter luciferase construct is inactive. But in Fig. 2d-2e and Fig. 3C, the authors show that T2D is associated with both reduced RHOT1 DNA methylation and lower expression of RHOT1. The two results are contradictory with each other.

2. New Fig. 1e, the authors only show DNA methylation levels of top 100 differential methylated sites. Although some sites do seem to be different from T2D and normal, target gene expression results are not shown in the figure. This is important because the entire paper is about linking DNA methylation and gene expression in T2D. Actually, I prefer the authors to show a figure for the "hits" (DEG with methylation changes) include gene names. The authors mentioned that they have 550 sites near 153 genes, which is not too many to be included in one figure. It will be more informative this way because at least readers will know if DNA hypermethylation is associated with up-or down-regulated genes in T2D (such as the RHOT1 example).

Response to review comments from Reviewer #3

Remarks to the Author:

I appreciate that the authors' efforts addressing my questions. However, I still have two major issues regarding the two newly added figures.

1. New Supp Fig. 5f. The authors shows that methylated RHOT1 promoter luciferase construct is inactive. But in Fig. 2d-2e and Fig. 3C, the authors show that T2D is associated with both reduced RHOT1 DNA methylation and lower expression of RHOT1. The two results are contradictory with each other.

Response:

Thanks for this comment, further helping us to improve the manuscript. We acknowledge that DNA methylation and its role in gene regulation is complex, and that we need to better explain why CpG sites in different gene regulatory regions, although within the same gene, cannot be assumed to have the same methylation levels or effects.

The CpG sites in **Fig. 2d-2e** and **Fig. 3c**, with lower methylation in islets from individuals with T2D, are within the gene body (intronic) of *RHOT1*. In contrast, the Luciferase experiment methylates all CpG sites including a CpG island in the promoter of *RHOT1*, 0-1540 bp upstream of transcription start site. The Luciferase assay methodology requires the CpG sites subject to methylation to be located within a gene region able to drive gene expression, i.e. promoter regions. Hence, the Luciferase experiment answers the fundamental question if DNA methylation may directly impact transcriptional activity / gene expression, based on your initial review comment regarding the mechanistic connection between DNA methylation and gene expression. Hence, the two results of *RHOT1* methylation, presented in **Fig. 2d-2e** and **Fig. 3c**, vs. **Figure S5f**, are independent findings and not contradictory.

In general, increased promoter DNA methylation is associated with decreased expression, e.g. through interfering with transcription factor binding, while intragenic/gene body methylation, is believed to enhance transcriptional elongation, hence having a positive association with gene expression. Thereby, methylation sites located in different gene regulatory regions, although within the same gene, may have opposing effects on gene transcription. We have previously shown this based on whole-genome bisulfite sequencing of human pancreatic islets (**Fig 1g**, Volkov et al., *Diabetes* 2017):

[REDACTED]
pmid: 28052964

We have now clarified in the manuscript that the intronic *RHOT1* methylation site associated with T2D is different from the sites methylated in the Luciferase experiment and why different levels of DNA methylation in different gene regulatory regions should not be considered contradictory results. Page 9, line 315-319:

“Of note, the promoter sites methylated in the Luciferase experiment are different from the intronic RHOT1 methylation site associated with T2D in human islets. CpG sites of different gene regulatory regions, although within the same gene, have different ways of influencing gene expression. For example, elevated promoter methylation has been shown to reduce expression, while intragenic methylation has been linked to higher expression (Ling et al 2022, Volkov et al 2017).”

Based on this comment, we also investigated the Epic array data in more detail, to find out if the *RHOT1* promoter region used for the Luciferase experiment is 1) covered by the Epic array, and 2) if yes, the DNA methylation status of this region based on array data in islets from donors with T2D versus controls. Although this 1540 bp *RHOT1* promoter region harbors 90 CpG sites, only nine were covered by the Epic array. The methylation level of the nine CpG sites located in the 1540 bp *RHOT1* promoter region was very low (~4%) with no significant difference between the T2D and control groups. It should be noted that probe design is difficult and less accurate for analysis of DNA methylation in CG rich regions such as CpG islands. Hence, based on this array analysis, we cannot exclude that T2D is associated with differential methylation of sites located in the *RHOT1* promoter. We added the following information on page 9, line 320-321 of the revised ms:

“Due to low Epic array coverage directly upstream of the RHOT1 TSS, we cannot conclude if T2D is associated with differential RHOT1 promoter methylation or not.”

2. New Fig. 1e, the authors only show DNA methylation levels of top 100 differential methylated sites. Although some sites do seem to be different from T2D and normal, target gene expression results are not shown in the figure. This is important because the entire paper is about linking DNA methylation and gene expression in T2D. Actually, I prefer the authors to show a figure for the “hits” (DEG with methylation changes) include gene names. The authors mentioned that they have 550 sites near 153 genes, which is not too many to be included in one figure. It will be more informative this way because at least readers will know if DNA hypermethylation is associated with up- or down-regulated genes in T2D (such as the *RHOT1* example).

Response:

This question is not easy to solve and comes back to the discussion above: different DNA methylation sites within the same gene may exhibit different levels of methylation and also have different ways of influencing gene expression, depending on the genomic location of the specific CpG sites. Simplifying epigenetic regulation to say that methylation of different gene regulatory features (promoters, enhancer regions, exon/intron boundaries, miRNA binding regions etc) connects to gene expression in the same way is therefore not appropriate, and hence it is not possible to give a general answer to if hypermethylation is associated with up- or down-regulation of a gene. Also, heatmaps show absolute values of individual samples, while you are asking for changes between groups (hypo- or hypermethylation). Although the information you ask for is not possible to visualize in a heatmap, the answer to if DNA hyper- or hypo-methylation is associated with up- or down-regulated genes in T2D for each specific CpG site is already provided in **Supplementary Table 9**. Here, it is also possible to see to what gene region the significant CpG sites are annotated to.

Additionally, to try to share some clarity based on this comment, we created a figure to visualize significant differences of DNA methylation in the promoter of DEGs in islet from T2D vs. control donors. This figure is based on the 153 genes differentially expressed in T2D that also show differential methylation, but instead of including all significant CpG sites annotated to those genes (which would be too many to include the gene names as asked for), we restricted the data set to CpG sites located within the promoter regions (TSS200 or TSS1500 based on Illumina annotation).

Please see new **Supplementary Fig 3g**, and the Results section, page 5, line 174-179:

“To visualize a possible link between concurrent differences in methylation and expression, we plotted relative differences of DNA methylation in the promoter of DEGs in islets of T2D cases versus controls (**Supplementary Fig. 3g, Supplementary Table 9**). Increased promoter methylation and decreased expression were seen for e.g. *BEST3*, *HHATL* and *SLC2A2*, while e.g. *IGF1*, *SFRP4* and *SYNPO* had reduced promoter methylation and increased expression in islets from T2D donors (**Supplementary Fig. 3g**).”

And the figure legend (page 32):

“**3 g**) Relative differences of promoter DNA methylation and gene expression in islets of T2D cases versus controls. Genes were selected based on differential expression in islets from T2D donors, and one or more differentially methylated CpG site(s) annotated to the promoter region (TSS200, TSS1500) of the same gene.”

Supplementary Fig. 3g shows that increased promoter methylation is often associated with decreased expression, while decreased methylation in several cases is associated with increased expression, as discussed in these two review comments. However, as mentioned, this is not always the case and the opposite can occur, as also seen in **Supplementary Fig. 3g**. Increased promoter methylation can result in decreased expression when DNA methylation sterically hinder transcription factor binding, or when methylation attracts methyl binding proteins that further recruit transcriptional repressors which closes down the chromatin structure (Ling et al., 2022). However, increased DNA methylation in promoter regions have also been found to increase expression, e.g., when DNA methylation takes place in sites where a transcriptional repressor bind. Based on this complexity, we added a short section in the discussion and refer to our review papers where we discuss this more.

Please see Discussion, page 12, line 434-438: *"It should be noted that the effect of DNA methylation depends on the exact location of the methylated site (Volkov et al., 2017, Ling et al., 2022). For example, promoter methylation is generally repressive while gene body methylation has been associated with increased gene expression. Two adjacent CpG sites can also have opposite effects, should they for example be located in the binding sites of a transcriptional activator or repressor, respectively."*

If this doesn't respond to your question, we would appreciate a clarification and help to interpret the issue. In the initial review comment, there was no mention of gene expression, hence we created a heatmap based on DNA methylation for each individual as asked for. *"I highly recommend including a panel in figure 1 (such as a heatmap) showing the changes the DNA methylation changes in every individual. This will help readers understand what kind of changes we are looking at, and how consistent the changes are between the control and T2D groups..."* Now, it is asked for a heatmap of DEGs, i.e., gene expression, *"I prefer the authors to show a figure for the "hits" (DEG with methylation changes)"*, whereas the next sentence ask for 550 methylation sites. We tried our best to respond, please let us know if there are more things to improve, or if you wanted the heatmap of RNA-Seq data.

In response to your initial comment, we choose to keep **Fig 1e** as it is, as it shows the absolute values of DNA methylation for each individual, and that the groups cluster based on methylation levels.